# COUNTERFACTUAL CONCEPT BOTTLENECK MODELS

**Gabriele Dominici**
Università della Svizzera italiana
gabriele.dominici@usi.ch

**Pietro Barbiero**
IBM Research*
pietro.barbiero@ibm.com

**Francesco Giannini**
Scuola Normale Superiore
francesco.giannini@sns.it

**Martin Gjoreski**
Università della Svizzera italiana
martin.gjoreski@usi.ch

**Giuseppe Marra**
KU Leuven
giuseppe.marra@kuleuven.be

**Marc Langheinrich**
Università della Svizzera italiana
marc.langheinrich@usi.ch

## ABSTRACT

Current deep learning models are not designed to simultaneously address three fundamental questions: *predict* class labels to solve a given classification task (the "What?"), *simulate* changes in the situation to evaluate how this impacts class predictions (the "How?"), and *imagine* how the scenario should change to result in different class predictions (the "Why not?"). While current approaches in causal representation learning and concept interpretability are designed to address some of these questions individually (such as Concept Bottleneck Models, which address both "what" and "how" questions), no current deep learning model is specifically built to answer all of them at the same time. To bridge this gap, we introduce CounterFactual Concept Bottleneck Models (CF-CBMs), a class of models designed to efficiently address the above queries all at once without the need to run post-hoc searches. Our experimental results demonstrate that CF-CBMs: achieve classification accuracy comparable to black-box models and existing CBMs ("What?"), rely on fewer important concepts leading to simpler explanations ("How?"), and produce interpretable, concept-based counterfactuals ("Why not?"). Additionally, we show that training the counterfactual generator jointly with the CBM leads to two key improvements: (i) it alters the model's decision-making process, making the model rely on fewer important concepts (leading to simpler explanations), and (ii) it significantly increases the causal effect of concept interventions on class predictions, making the model more responsive to these changes.

## 1 INTRODUCTION

To calibrate human trust and enhance human-machine interactions, deep learning (DL) models should learn how to master three fundamental questions: *predict* class labels for new inputs (the "What is the diagnosis for the X-ray shown in Figure 1?"), *simulate* changes in the situation to evaluate how this impacts class predictions (the "How would the absence of collapsed lung impact the presence of pneumothorax?"), and *imagine* al-

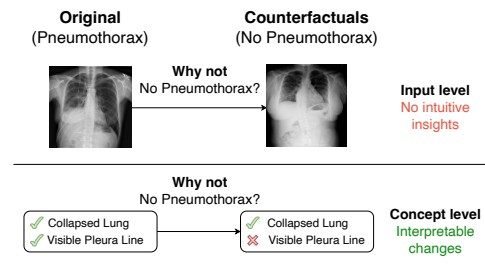

Figure 1: CF-CBMs generate counterfactuals at the concept level rather than at the input level, as the changes are more interpretable. On the contrary, identifying the changes in the input level require significant more effort from the user. The images are sourced from the SIIM Pneumothorax dataset (Zawacki et al., 2019).

---

*Work conducted while employed at Università della Svizzera italiana.

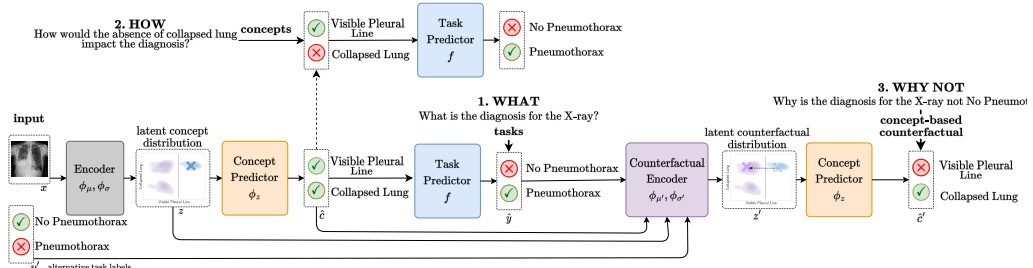

Figure 2: Counterfactual CBM. For a given input sample, the task predictor answers "what?" queries predicting class labels. The concept predictor answers "how?" queries simulating changes in the scenarios through interventions. When taking the counterfactual latent distribution $z'$, the concept predictor answers "why not?" queries via concept-based counterfactuals, which are easier to understand then counterfactuals at the input level.

ternative scenarios that would result in different class predictions (the "Why is the patient *not having pneumothorax*?")—also known as counterfactual explanations (Wachter et al., 2017). Currently, no existing DL model is designed to answer all these questions by design.

Concept Bottleneck Models (CBMs) (Koh et al., 2020) are DL architectures designed to answer both "what?" and "how?" questions. They first predict human-understandable concepts (Kim et al., 2018), allowing experts to evaluate how intervening on such concepts influences the model's class predictions. However, CBMs fall short in answering "why not?" queries as they are not trained to produce concept-based counterfactuals. More broadly, counterfactuals have been explored in concept-based models, e.g., for concept discovery (Stammer et al., 2024), while causal inference addresses this statistically (Louizos et al., 2017; Xia et al., 2022), seeking to discover and/or exploit the true causal relationships in the data. In contrast, we focus on addressing these questions from the model's perspective, shedding light on its decision-making process. From this viewpoint, state-of-the-art counterfactual generation methods (Wachter et al., 2017; Pawelczyk et al., 2020; Guyomard et al., 2023; Nemirovsky et al., 2021) answer "what?" and "why not?" but not "how?", especially when working with unstructured data such as images. Moreover, in these cases, counterfactual methods may generate an image that either i) looks nearly identical to the original, or ii) incorporates pixel-level modifications of difficult interpretation from the human perspective across thousands of pixels throughout the entire image (as shown in Figure 1). This scattering of modifications makes it hard to understand the answers to the "why not" questions. In addition, it is difficult to answer "how" questions because semantically coherent pixel-level interventions are hard to define and perform for domain experts (e.g., radiologists) (Abid et al., 2022). In summary, input-level counterfactuals often lack actionable insights, while concept-based models cannot answer counterfactual queries.

**Contributions.** To bridge these gaps, we introduce CounterFactual Concept Bottleneck Models (CF-CBM), a class of deep learning models designed to jointly master three fundamental questions for a classification problem - "What?", "How?" and "Why not?", generating interpretable concept-based counterfactuals. Our key innovation is a latent process that generates two similar concept vectors: one predicts the target class label (as in standard CBMs), while the other predicts an alternative class. Our experiments show that CF-CBMs achieve classification accuracy comparable to black-box models and existing CBMs ("What?"), rely on fewer important concepts leading to simpler explanations ("How?"), and produce interpretable, concept-based counterfactuals ("Why not?"). Additionally, training the counterfactual generator jointly with the CBM leads to two key benefits: (i) it alters the model's decision-making, making it rely on fewer important concepts and (ii) it significantly increases the causal effect of concept interventions on class predictions, making the model more responsive to these changes. The code of this paper is publicly available[1].

## 2 BACKGROUND

**Concept Bottleneck Models (CBMs).** Concept Bottleneck Models (Koh et al., 2020) are interpretable architectures that explain their predictions using high-level units of information (i.e., "concepts").

---

[1]https://github.com/gabriele-dominici/Counterfactual-CBM

Given a sample's raw features $x \in X \subseteq \mathbb{R}^d$ (e.g., an image's pixels), a set of $r$ concepts $c_i \in C \subseteq \{0,1\}^r$ (e.g., "collapsed lung", "visible pleural line"), and a set of $l$ class labels $y_j \in Y \subseteq \{0,1\}^l$ (e.g., labels "pneumothorax" or "no pneumothorax"). A CBM estimates the conditional distribution $\prod_j p(y_j \mid c_1, \ldots, c_r) \prod_i p(c_i \mid x)$ where $p(c_i \mid x)$ is a set of independent Bernoulli distributions. At test time, human experts may *intervene* on mispredicted concept labels to improve CBMs' task performance, simulating different scenarios. In a probabilistic perspective (Bahadori & Heckerman, 2020; Misino et al., 2022), (ideal) concepts $c$ represent key factors of variation (Kingma & Welling, 2013; Zarlenga et al., 2023) of observed data $x$ and $y$, as illustrated in the graphical model $\boxed{x} \leftarrow \boxed{c} \rightarrow \boxed{y}$ .

**Counterfactual Explanations.** A counterfactual is a hypothetical statement in contrast with actual events that helps us understand the potential consequences of different choices or circumstances (Pearl et al., 2016). In the context of machine learning, Wachter et al. (2017) define counterfactual explanations as an optimization problem representing an answer to a "why not" question. The optimization aims to find for each observation $x$ the closest datapoint $x'$ such that a classifier $m : X \to Y$ produces a class label $m(x')$ that is different from the original label $m(x)$.

## 3 COUNTERFACTUAL CONCEPT BOTTLENECK MODELS

CounterFactual Concept Bottleneck Models (CF-CBMs, Figure 2) is a novel class of interpretable models designed to generate counterfactuals via variational inference at the concept level. In the following, we motivate CF-CBMs' architecture and optimization objective (Section 3.1). Sections 3.2 and 3.3 show how CF-CBMs answer the different questions and how users can act upon generated counterfactuals.

### 3.1 GENERATING CONCEPT-BASED COUNTERFACTUALS

To realize concept-based counterfactuals, we extend the graphical model $\boxed{x} \leftarrow \boxed{c} \rightarrow \boxed{y}$ introducing two additional variables: $c'$, representing concept-based counterfactual labels; and $y'$, representing the counterfactual class label. We represent the counterfactual dependency on the actual concepts $c$ with an arrow from $c$ to $c'$, while $y'$ depends on $c'$:

$$
\begin{array}{c}
\boxed{c'} \rightarrow \boxed{y'} \\
| \\
\boxed{x} \leftarrow \boxed{c} \rightarrow \boxed{y}
\end{array}
\tag{1}
$$

We also notice that multiple counterfactuals can be used to explain any given fact (Wachter et al., 2017; Pearl, 2000). To address this, an ideal solution would be to use a generative approach modeling the discrete distributions over $c$ and $c'$. However, modeling such distributions directly might be unfeasible in practice (Richardson & Domingos, 2006) as it requires either modelling complex concept dependencies with discrete distributions which scale exponentially with the number of concepts, or assuming that concepts are independent of each other, an often unrealistic assumption. For this reason, we use a latent variable approach (Kingma & Welling, 2013) to model concept dependencies in a continuous latent distribution, which allows us to model discrete concept values as independent of each other given the latent variable.

**Architecture.** CF-CBMs are latent variable models generating counterfactuals via variational inference. To this end, they add two random variables $z$ and $z'$ to Diagram 1. These variables represent latent factors of variation whose probability distributions are easier to model and sample compared to those for $c$ and $c'$. We also include arrows from $c$ and $y$ to the counterfactual latent distribution $z'$ in order to explicitly model the dependency of $z'$ on the symbolic values of actual concept $c$ and class labels $y$, resulting in the following overall probabilistic graphical model:

$$
\begin{array}{c}
\boxed{z'} \leftarrow \boxed{c'} \rightarrow \boxed{y'} \\
\boxed{x} \leftarrow \boxed{z} \rightarrow \boxed{c} \rightarrow \boxed{y}
\end{array}
\tag{2}
$$

This way, the generative distribution factorizes as:

$$
p(c, y, z, c', y', z', x) = p(c, y|z)p(c', y'|z')p(x|z)p(\mathbf{z}|c, y)
\tag{3}
$$

$$
p(c, y|z) = p(y|c)p(c|z), \quad p(c', y'|z') = p(y'|c')p(c'|z'), \quad p(\mathbf{z}|c, y) = p(z)p(z'|z, c, y)
\tag{4}
$$

In our approach, $p(y|c)$ and $p(y'|c')$ are modeled as categorical distributions parametrized by the same task predictor $f$; $p(c|z)$ and $p(c'|z')$ as sets of Bernoulli distributions parametrized by the same concept predictor $\phi_z$. In practice, we assume that the input x is always observed at test time, making the p(x|z) term irrelevant. This allows $z$ and $z'$ to encode only the most salient information about the concepts c and c', rather than all the details needed to reconstruct x. As a result, the optimization process is significantly simplified, enabling efficient counterfactual generation for complex data like images. Finally, $p(z)$ is a normal prior distribution and $p(z'|z, c, y)$ is a learnable normal prior whose mean and variance are parametrized by neural networks $\phi_{p\mu}$ and $\phi_{p\sigma}$. The added complexity introduced with these additional black-box components is limited to the stages before concept extraction, enabling the encoding of information about possible counterfactuals. Thus, our model retains the interpretability of CBMs while providing the additional capability to explain decisions through counterfactuals.

**Amortized inference.** CF-CBMs amortize inference needed for training by introducing two approximate Gaussian posteriors $q(z|x)$ and $q(z'|z, c, y, y')$ whose mean and variance are parametrized by a pair of neural networks $(\phi_\mu, \phi_\sigma)$ $((\phi_{\mu'}, \phi_{\sigma'})$, respectively). The corresponding inference graphical model (i.e. the encoder) is shown in Appendix A.

**Optimization problem.** CF-CBMs are trained to optimize the log-likelihood of tuples $(c, y, y')$, where $y'$ is randomly sampled from a uniform distribution for each observation of $x$, simulating a scenario in which a user requests the generation of a counterfactual for a specific label. Following a variational inference approach, we optimize the evidence lower bound of the log-likelihood, which results (see Appendix A) in the following objective function to maximize:

$$\mathcal{L} = \lambda_1 \overbrace{\mathbb{E}_{z \sim q(z|x)}[\log p(c|z)] + \log p(y|c)}^{\text{reconstruction of } c \text{ and } y} - \lambda_2 \overbrace{D_{KL}[q(z|x)||p(z)]}^{\text{prior regularization on } z}$$

$$+ \lambda_3 \overbrace{\mathbb{E}_{z, z', c' \sim p(c'|z')q(z'|\alpha)q(z|x)}[\log p(y'|c')]}^{\text{reconstruction of } y'} - \lambda_4 \overbrace{D_{KL}[q(z'|\alpha)||p(z'|z, c, y)]}^{\text{prior regularization on } z'}$$

where $D_{KL}$ is the Kullback–Leibler divergence, $\alpha = (z, c, y, y')$ and $\lambda_i$ are hyperparameters. Moreover, in order to enforce concept-based counterfactuals to be as close as possible to the current concept labels, we add an additional term to the objective (Kingma & Welling, 2022):

$$\mathcal{L}_{dz} = - \lambda_5 \overbrace{D_{KL}[q(z|x)||q(z'|\alpha)]}^{\text{posterior distance}} - \lambda_6 \overbrace{D_{KL}[p(z)||p(z'|z, c, y)]}^{\text{prior distance}} \quad (5)$$

## 3.2 ANSWERING "WHAT?", "HOW?" AND "WHY NOT?" QUERIES

CF-CBMs design allows them to answer "what?", "how?" and "why not?" queries through the following steps (Figure 2):

**1. What? Predict** concept and class labels:

  (a) **Sample from latent concept posterior**: $z \sim \mathcal{N}(\phi_\mu(x), \phi_\sigma(x))$
  (b) **Sample to predict concept and class labels**: $\hat{y} \sim \text{Cat}(f(\hat{c})), \ \hat{c} \sim \text{Ber}(\phi_z(z))$

**2. How? Simulate** changes in the situation and evaluate the impact on the outcome. Similar to standard CBMs, it is possible to intervene at the concept level—since these concepts are inherently interpretable—and analyze how these modifications impact the model's prediction:

  (a) **Simulate changes intervening on concept values**: $\hat{c}_i := \tilde{c}_i \quad \forall i \in \mathcal{I}$ where $\tilde{c}_i$ is the intervened value for each concept $i$ in the set $\mathcal{I}$ of the intervened concepts.
  (b) **Sample to predict class labels based on the intervened concepts**: $\tilde{y} \sim \text{Cat}(f(\hat{c}))$

**3. Why not? Imagine** interpretable counterfactuals via the generation of new concept tuples:

  (a) **Sample from latent counterfactual posterior**:
  $$z' \sim \mathcal{N}(\phi_{\mu'}(\alpha), \phi_{\sigma'}(\alpha)), \quad \alpha = (z, \hat{c}, \hat{y}, y'), \quad y' \sim \text{Categorical}(v), \ v \sim \mathcal{U}\{0, |y|\}$$
  (b) **Sample to predict counterfactual labels**: $\hat{y}' \sim \text{Cat}(f(\hat{c}')), \ \hat{c}' \sim \text{Ber}(\phi_z(z'))$

The following example illustrates a concrete scenario.

**Example.** Consider a lung X-ray scan with the task of classifying if a patient has pneumothorax or not. Additionally, we have access to two key concepts: "collapsed lung" and "visible pleural line", which are critical in determining if a patient has pneumothorax or not. A CF-CBM addresses the three interpretability questions as follows: *Predict*: the model predicts concept labels $\hat{c} = \hat{c}_{CollapsedLung} = 1, \hat{c}_{VisiblePleuralLine} = 1$ and class labels $\hat{y}_{NoPneumothorax} = 0, \hat{y}_{Pneumothorax} = 1$, indicating the presence of both concepts and a Pneumothorax classification. *Simulate*: it is possible to intervene on the concept "collapsed lung" $\hat{c}_{CollapsedLung} = 0$, simulating a different scenario, and observe that the model still predicts the same class, presence of Pneumothorax, showing the model's robustness to changes in one concept. *Imagine*: generate the counterfactual where the desired class changes to $\hat{y}'_{NoPneumothorax} = 1, \hat{y}'_{Pneumothorax} = 0$. The corresponding concept-level counterfactual would be $\hat{c}'_{CollapsedLung} = 1, \hat{c}'_{VisiblePleuralLine} = 0$, demonstrating how modifying these concepts could lead to a different classification.

### 3.3 TEST-TIME FUNCTIONALITIES

In contrast to standard CBMs, CF-CBMs can either sample or estimate the most probable counterfactual to: (i) explain the effect of concept interventions on tasks, (ii) show users how to get a desired class label, and (iii) propose concept interventions via "task-driven" interventions.

**Counterfactuals explain the effect of concept interventions on downstream tasks.** Plain concept-based explanations indicate the presence or absence of a concept for a given class prediction. However, the complexity of plain explanations grows quickly with the number of concepts. Concept-based counterfactuals, instead, induce simpler, sparser explanations (via Eq. 5) representing minimal modifications of concept labels that would have led to a different class prediction.

**Act upon concept-based counterfactuals.** Concept-based counterfactuals can guide users towards achieving desired outcomes (indicated by class labels), especially when users do not know how to alter their status (represented by concepts). In this scenario: (i) a user specifies a desired class label $y'$ for the downstream task representing their goal (e.g., get a loan), (ii) CF-CBMs generate a concept-based counterfactual $p(c'|y', \hat{y}, \hat{c})$ representing minimal concept modifications changing the class label from the predicted $\hat{y}$ to the desired $y'$ (e.g., save 10% more each month), (iii) the user can act upon the counterfactual $c'$ to accomplish the goal represented by $y'$.

**Task-driven interventions fix mispredicted concepts.** A key feature of CBMs is enabling human-in-the-loop interventions: at test time, users can correct mispredicted concept labels to enhance downstream task performance ($c \rightarrow y$ intervention). Concept interventions can be useful when intervening on concepts is easier than intervening on the downstream task. For instance, it might be easier to identify if the lung is collapsed ($c$) in a lung X-ray scan than to provide an accurate prediction if a patient has or not pneumothorax ($y$). While still supporting concept interventions, CF-CBMs may also invert this mechanism via *task-driven interventions* ($y \rightarrow c$), exploiting its counterfactual generation abilities. Task-driven interventions can be used to correct mispredicted class labels when intervening on the downstream task is easier than intervening on concepts. For instance, it might be easier to identify a mispredicted Parkinson's disease ($y$) rather than intervening in the genetic pathways ($c$) leading to the disease. In this scenario, users can intervene on the class label (rather than on the concepts) suggesting a correct $y'$. Considering this additional information, CF-CBMs propose a more accurate set of concept labels $\hat{c}'$ representing potential concept intervention previously unknown to the user.

## 4 EXPERIMENTS

Our experiments aim to answer the following questions:

- **What? prediction generalization:** Do CF-CBMs attain similar task/concept accuracy as standard CBMs?

- **How? intervention impact:** Does optimizing counterfactual generation end-to-end with the predictor alter the importance assigned by the model to each concept?

- **Why not? counterfactual actionability:** Can CF-CBMs produce valid, plausible, efficient counterfactuals? Can CF-CBMs generate accurate task-driven interventions?

This section describes essential information about experiments. We provide further details in Appendix C.

**Data & task setup**  In our experiments we use five different datasets commonly used to evaluate CBMs: dSprites (Matthey et al., 2017), where the task is to predict specific combinations of objects having different shapes, positions, sizes and colours; MNIST addition (Manhaeve et al., 2018), where the task is to predict the sum of two digits; and CUB (Welinder et al., 2010), where the task is to predict bird species based on bird characteristics; CIFAR10 (Krizhevsky et al.), where the task is to classify the object in the image, and SIIM Pneumothorax (Zawacki et al., 2019), where the task is to determine whether the X-ray scan indicates Pneumothorax. These two datasets do not include concept annotations, so we extract them following the method proposed in Oikarinen et al. (2023). In all experiments, images are encoded using a pre-trained ResNet18 (He et al., 2015) (dSpties, MNIST Add, CUB), CLIP ViTB16 (Radford et al., 2021)(CIFAR10) or CXR-CLIP (You et al., 2023) (SIIM Pneumothorax).

**Evaluation**  In our analysis we use the following metrics. **What? (prediction generalization)**: we compute the Area Under the Receiver Operating Characteristic Curve (Hand & Till, 2001) for concepts and tasks ($ROC\ AUC$ ($\uparrow$)[2]). **How? (intervention impact)**: we assess the influence of concepts on the downstream task using the Causal Concept Effect (CaCE) (Goyal et al., 2019), which quantifies the degree to which each concept is critical for a class. This allows us to evaluate the impact of optimizing counterfactuals end-to-end versus using post-hoc methods. Additionally, in scenarios where confounders are present, CaCE can help determine whether the impact of these confounders is high or low. **Why not? (counterfactual actionability)**: drawing from previous works on counterfactuals, we compute: (i) the **validity**[3]($\uparrow$) (Wachter et al., 2017) by checking whether the model predicts the desired class labels based on the generated counterfactual; (ii) the **proximity** ($\downarrow$) (Pawelczyk et al., 2020) which evaluates counterfactuals' "reliability" as their similarity w.r.t. training samples; (iii) the **time** ($\downarrow$) (Romashov et al., 2022) to generate counterfactuals (see Appendix D for the results on this metric). In addition, we propose the following metrics to assess the quality of counterfactuals: (i) the $\Delta$-**Sparsity** ($\downarrow$) which evaluates user's "wasted efforts" by counting the number concepts changed (Guo et al., 2023) w.r.t. the minimal number of changes that would have generated a counterfactual according to the dataset; (ii) the "plausibility" (Wachter et al., 2017) as the **Intersection over Union** ($IoU$ ($\uparrow$)) (Jaccard, 1912) between counterfactuals and ground-truth concept vectors; (iii) the **variability** ($\uparrow$) which evaluates counterfactuals' "diversity", as the cardinality ratio between the set of counterfactuals generated and the set of training concept vectors; (iv) finally, we measure the **concept accuracy of generated interventions** ($Acc\ Int.$ ($\uparrow$)) w.r.t. ground-truth optimal interventions that evaluate the model's ability to generate potential concept interventions. Following Espinosa Zarlenga et al. (2022), we inject noise on predicted concepts to reduce task accuracy, then we sample counterfactuals, conditioning on the ground-truth label, and use them to fix mispredicted concept labels automatically. All metrics are reported using the mean and the standard error over five different runs with different initializations. Additionally, in Appendix E, we provide examples of counterfactuals generated by our models and the baselines across various datasets.

**Baselines**  In our experiments, we compare CF-CBM with a Black Box model, a standard CBM and a more powerful CBM as Concept Embedding Model (CEM) (Espinosa Zarlenga et al., 2022) in terms of generalization performance. To ensure a fair comparison, we create counterfactual baselines by adapting post-hoc methods to standard CBMs, allowing for a direct comparison between our approach and other state-of-the-art models. We propose this experimental design for two main reasons. First, standard CBMs cannot generate counterfactuals by design, hence they need an external generator. Second, the standard, direct application of post-hoc methods on unstructured data types (e.g., image pixels rather than concepts) would have significantly compromised counterfactuals' interpretability and difficult comparison between the generated results. We remark that this already represents an undocumented procedure in the current literature, as post-hoc methods typically derive counterfactuals from input features, not in the concept space. However, this also enables us to compare post-hoc

---

[2]($\uparrow$): the higher the better. ($\downarrow$): the lower the better.
[3]A.k.a. "correctness" (Abid et al., 2022).

counterfactual methods with our model, which incorporates counterfactual constraints during training. Among post-hoc approaches we select: (i) the Bayesian Counterfactual (BayCon) (Romashov et al., 2022), which generates counterfactuals via probabilistic feature sampling and Bayesian optimization; (ii) the Counterfactual Conditional Heterogeneous Variational AutoEncoder (C-CHVAE) (Pawelczyk et al., 2020), which iteratively perturbs the latent space of a VAE until it finds counterfactuals; (iii) the VAE CounterFactual (VAE-CF) (Mahajan et al., 2020), which improves C-CHVAE by conditioning counterfactuals on model's predictions; (iv) the Variational Counter Net (VCNet) (Guyomard et al., 2023), which conditions latent counterfactual sampling on a specific target label $y'$.

## 5 KEY FINDINGS & RESULTS

### 5.1 WHAT? PREDICTION GENERALIZATION

**CF-CBMs achieve generalization performance that is close to standard CBMs.** CF-CBMs attain both concept and task ROC AUC similar to Black Box, standard CBM and CEM, a more expressive CBM (details in Appendix D). This shows that generating counterfactuals does not have a negative impact on classification performance on the datasets we considered. Our goal was not to improve performance but to improve interpretability. This outcome highlights how CF-CBMs generalise standard CBM architectures by matching their predictive accuracy while introducing new fundamental functionalities (i.e., counterfactual explanations and task-driven interventions).

### 5.2 HOW? INTERVENTION IMPACT

**Jointly training counterfactuals' generator and CBMs makes the model rely on fewer important concepts leading to simpler explanations (Table 1)** Following Koh et al. (2020), CF-CBMs allow users to intervene directly on the concepts, enabling simulation of diverse scenarios and observation of their effects on the model's final prediction, as quantified by the Causal Concept Effect (CaCE) (Goyal et al., 2019). To assess the impact of CF-CBMs joint training design on how the model perceives the importance of each concept we conducted experiments using a modified dSprites dataset, following Goyal et al. (2019), where a high correlation was introduced between an object's color and its class label. While standard CBM explanations weighted shape and color (confounders) equally for class prediction, CF-CBMs correctly prioritized shape as the primary predictor, as shown in

Table 1: **CF-CBMs ignore the confounding concept** on the modified dSprites dataset.

|  | Post-hoc | Joint (ours) |
| --- | --- | --- |
| Shape 1 | $0.16 \pm 0.03$ | $0.30 \pm 0.15$ |
| Shape 2 | $0.13 \pm 0.03$ | $0.18 \pm 0.08$ |
| Shape 3 | $0.16 \pm 0.03$ | $0.32 \pm 0.09$ |
| Two obj | $0.04 \pm 0.01$ | $0.16 \pm 0.07$ |
| Colours (confounder) | $0.17 \pm 0.03$ | $0.02 \pm 0.01$ |

Table 2: **Compared to CBMs, CF-CBMs assign greater importance to a small fraction of concepts, while the majority of concepts receive less importance** on CUB.

| Dataset | Percentile | Mean Post-hoc | Mean Joint (ours) | p-value |
| --- | --- | --- | --- | --- |
| MNIST | Top 10% (↑) | $0.1589 \pm 0.0203$ | $\mathbf{0.1827} \pm 0.0676$ | **0.0001** |
|  | Bottom 50% (↓) | $0.0308 \pm 0.0175$ | $\mathbf{0.0257} \pm 0.0143$ | 0.2437 |
| CUB | Top 10% (↑) | $0.0093 \pm 0.0029$ | $\mathbf{0.0131} \pm 0.0044$ | **0.0** |
|  | Bottom 50% (↓) | $0.0016 \pm 0.0006$ | $\mathbf{0.0013} \pm 0.0008$ | **0.0** |

Table 1. Specifically, altering color in standard CBMs resulted in a CaCE score of 0.17, demonstrating a 17% chance of prediction change due to this confounding factor. This reveals that standard CBMs are susceptible to relying on shortcuts in decision-making (Marconato et al., 2023). In contrast, CF-CBMs showed minimal sensitivity to color changes, with a CaCE score of just 0.02, demonstrating significantly higher robustness against confounding features. Additionally, CF-CBMs assigned higher CaCE scores to shape concepts, indicating a stronger reliance on these critical features, unlike the post-hoc version. This trend was consistently observed across the other datasets: MNIST (p-value = 0.003) and CUB (p-value = 0.000), where CF-CBMs and CBMs exhibited distinct CaCE distributions as confirmed by the Kolmogorov-Smirnov test (the distributions are shown in App. D). Table 2 further illustrates that CF-CBMs achieved a higher mean CaCE for the top 10% of concepts, signifying that the model concentrates on a smaller subset of highly relevant concepts. This focus results in more concise and interpretable explanations by engaging a reduced number of concepts. Moreover, CF-CBMs exhibited a lower mean CaCE for the bottom 50% of concepts, suggesting that the model learned to disregard less critical features, *thereby enhancing robustness against concept perturbations*. The Mann-Whitney U test results reveal that three out of four comparisons were statistically significant, underscoring the impact of joint training. This shift in decision-making comes from the CF-CBM's

Table 3: **CF-CBMs outperform all baselines in counterfactuals' validity** across all datasets.

| | dSprites (↑) | MNIST add (↑) | CUB (↑) | CIFAR10 (↑) | SIIM Pneumothorax (↑) | avg. (↑) |
|---|---|---|---|---|---|---|
| CBM+BayCon | $100.0 \pm 0.0$ | $94.6 \pm 0.5$ | $66.2 \pm 0.5$ | $91.0 \pm 3.3$ | $100.0 \pm 0.0$ | $90.4 \pm 0.7$ |
| CBM+CCHVAE | $98.4 \pm 0.6$ | $94.7 \pm 0.2$ | $78.7 \pm 0.4$ | $93.6 \pm 0.1$ | $93.0 \pm 0.6$ | $91.7 \pm 0.2$ |
| CBM+VAECF | $100.0 \pm 0.0$ | $82.0 \pm 5.9$ | $95.2 \pm 0.2$ | $91.0 \pm 0.8$ | $100.0 \pm 0.0$ | $93.4 \pm 1.2$ |
| CBM+VCNET | $100.0 \pm 0.0$ | $89.9 \pm 0.9$ | $93.1 \pm 0.5$ | $100.0 \pm 0.0$ | $100.0 \pm 0.0$ | $96.6 \pm 0.2$ |
| CF-CBM (ours) | $100.0 \pm 0.0$ | $96.4 \pm 0.7$ | $99.0 \pm 0.1$ | $100.0 \pm 0.0$ | $99.9 \pm 0.1$ | $99.06 \pm 0.1$ |

objective of generating counterfactuals that alter only the minimal subset of concept labels required to change the class prediction—e.g., adjusting shape but not color. Consequently, the model's task predictor is forced to focus on the minimal set of decisive features. For additional details on the CaCE score distributions, refer to Appendix D. Overall, our findings suggest that jointly training CF-CBMs not only improves model interpretability but also yields a more robust decision-making process.

### 5.3 WHY NOT? COUNTERFACTUAL ACTIONABILITY

**CF-CBMs outperform baselines in counterfactuals' validity. (Table 3)** CF-CBMs attain the highest counterfactual validity across all datasets when compared to counterfactual generation baselines applied to the concept space of standard CBMs. This result shows that CF-CBMs can effectively generate concept vectors that lead to user-requested class labels. The gap w.r.t. baselines increases in complex datasets involving a vast search space over multiple concepts and classes. In the MNIST addition dataset (20 concepts, 19 classes), CF-CBMs achieve +14 percentage points (p.p.) higher validity than VAECF and +7 p.p. higher than VCNet. In CUB (118 concepts, 200 classes), CF-CBMs find solutions with a significantly higher validity w.r.t. iterative methods: +33 p.p. higher than BayCon and +20 p.p. higher than C-CHVAE. Lastly, in the CIFAR10 and SIIM Pneumothorax datasets, CF-CBMs continue to achieve the highest validity.

**CF-CBMs balance counterfactuals' reliability with minimal user effort. (Figure 3)** High-quality counterfactuals should satisfy two key properties simultaneously: to be reliable and to require the least amount of users' effort to act upon. However, minimizing user's effort alone might lead to learning shortcuts producing less reliable counterfactuals, while optimizing reliability might generate counterfactuals that change the values of multiple concepts and represent a significant burden for users to take actions. For this reason, these two properties should be evaluated together:

- **CF-CBMs generate reliable counterfactuals:** When processing large datasets, the training distribution covers a large portion of the feasible action space. For this reason, counterfactual solutions close to training samples are likely to be more reliable, as discussed by Pawelczyk et al. (2020); Laugel et al. (2019); Wachter et al. (2017). Our results show that CF-CBMs generate highly reliable counterfactuals across all datasets. Baseline models instead struggle to produce counterfactual close to training samples in at least one dataset.

- **CF-CBMs generate counterfactuals requiring minimal user effort to act upon:** CF-CBMs consistently generate counterfactuals close to the optimal sparsity i.e., the fewest possible concept changes to alter the class prediction. This represents a significant result in terms of actionability because each concept modification requires user effort. For this reason, should a counterfactual require many concept label changes compared to the original concept vector, it would lead to a substantial unnecessary effort. Across all datasets, CF-CBMs stand out for their consistency in this aspect: their counterfactuals are either the sparsest or very close to the sparsest. This makes them highly effective and user-friendly, avoiding the burden of unnecessary actions.

**CF-CBMs calibrate the variability-plausibility trade-off. (Figure 4)** Another key trade-off in generating counterfactuals corresponds to the tension between producing counterfactuals with high diversity, allowing users to pick among different roadmaps to action, and producing plausible counterfactuals corresponding to user actions that are likely to be feasible in practice. However, learning shortcuts might lead to trivial solutions with a poor variability-plausibility trade-off. For instance, once a plausible counterfactual is found, a trivial solution could be to generate always the same counterfacual, thus leading to minimal diversity, while maximizing the variability increases the chance of generating less plausible counterfactuals. In practice, these two properties are measured by variability and IoU between counterfactuals and ground-truth concept vectors. A higher variability implies that our model can produce a wider range of counterfactuals. A higher IoU value suggests that

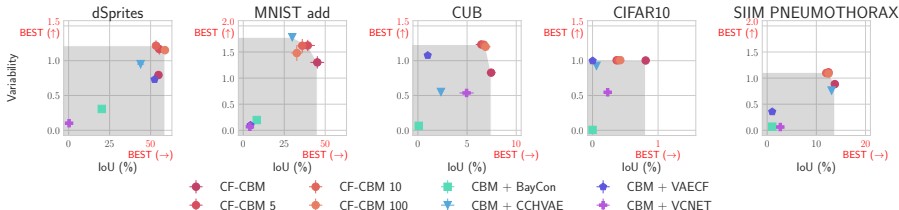

Figure 3: **CF-CBMs balance the trade off between counterfactuals' reliability** (proximity) **and user effort** ($\Delta$-Sparsity). The arrow ($\rightarrow$) points towards optimal values. Pareto-optimal models form the frontier of the shaded region, whereas dominated solutions are located within the shaded region.

Figure 4: **CF-CBMs calibrate the trade off between counterfactuals' diversity** (variability) **and plausibility** (IoU). The arrow ($\rightarrow$) points towards optimal values. Pareto-optimal models form the frontier of the shaded region, whereas dominated solutions are located within the shaded region. "CF-CBM $n$" samples $n$ counterfactuals from the latent distribution as described in App. B.

generated counterfactuals are also representative of the training distribution which contains plausible solutions. Our results show that CF-CBMs balance the optimization of these two properties better than existing methods. The gap w.r.t. BayCon and VCNet is already significant on dSprites where CF-CBMs produce counterfactuals with $+25$ p.p. higher IoU (plausibility) and $\times 3$ more variable. In MNIST addition the VAECF's performances significantly deteriorate, while C-CHVAE mainly struggles in CUB. On CIFAR10, all baselines perform poorly on the IoU metric, while on the SIIM Pneumothorax dataset, they also struggle with variability, with the exception of C-CHVAE.

**CF-CBMs generate accurate task-driven interventions. (Figure 5)** CF-CBMs enable task-driven interventions, showing users how to modify concepts to get the desired class label. This represents a novel crucial feature when users know the desired outcome but lack the knowledge to intervene on concepts. CF-CBM significantly outperforms other baselines in concept accuracy of task-driven interventions by up to 78 p.p. Such performance can be explained as the generation of task-driven interventions is conditioned on a user-provided target class label which provides fundamental information for the model to search in the latent concept distribution. Appendix D shows that CF-CBMs attain similar results while applying different levels of noise on predicted concept labels.

## 6 DISCUSSION & CONCLUSION

**Limitations.** CF-CBMs combine features from CBMs and generative models, carrying forward their strengths and limitations. From generative models, CF-CBMs inherit the ability to efficiently approximate the data distribution. However, this approximation does not guarantee counterfactuals' optimality when compared to exact methods (Wachter et al., 2017). Besides, CF-CBMs require concept annotators (humans or machines (Oikarinen et al., 2023)) to ground explanations, as standard CBMs. Reasoning shortcuts (Marconato et al., 2023), concept impurities (Zarlenga et al., 2023), and information bottlenecks (Espinosa Zarlenga et al., 2022) are also typical limitations of CBM architectures, especially when the concept bottleneck is not complete (Yeh et al., 2020).

**Relations with concept-based models.** CF-CBMs share common ground with existing CBMs in modeling concepts in latent spaces. However, current models use the latent concept space to improve task accuracy (Espinosa Zarlenga et al., 2022; Barbiero et al., 2023; Debot et al., 2024), promote concept disentanglement (Misino et al., 2022; Marconato et al., 2022), model concept interactions (Xu et al., 2024; Barbiero et al., 2024), or estimate task confidence intervals (Kim et al., 2023). Differently,

Figure 5: **CF-CBMs generate reliable** (proximity) **and accurate interventions** (ROC AUC Int.). The arrow ($\rightarrow$) points towards optimal values. Pareto-optimal models form the frontier of the shaded region, whereas dominated solutions are located within the shaded region.

Table 4: CF-CBMs efficiently address "what?", "how?", and "why not?" questions while supporting both concept and task-interventions, as opposed to existing CBMs and counterfactual generators.

|  | **CF-CBM (ours)** | CBM | Black Box | BayCon | C-CHVAE | VAE-CF | VCNet |
|---|---|---|---|---|---|---|---|
| What? | ✓ | ✓ | ✓ |  |  |  | ✓ |
| How? | ✓ | ✓ |  |  |  |  |  |
| Why not? | ✓ |  |  | ✓ | ✓ | ✓ | ✓ |
| Concept interventions | ✓ | ✓ |  |  |  |  |  |
| Task-driven interventions | ✓ |  |  |  |  |  |  |
| Real-time | ✓ | ✓ |  |  |  | ✓ | ✓ |

CF-CBMs leverage the latent concept space to extend CBMs' capabilities by introducing two key functionalities (Table 4): counterfactual explanations and task-driven interventions.

**Relations with generative structured models.** CF-CBMs are latent generative models characterized by a structured latent space. This characterization aligns them with other methodologies utilizing hierarchical priors (Sønderby et al. (2016); Maaløe et al. (2019); Vahdat & Kautz (2020); Klushyn et al. (2019)), mixture priors (Bauer & Mnih (2019)), and autoregressive priors (Chen et al. (2017); van den Oord et al. (2017); Razavi et al. (2019)). However, the structure of CF-CBMs' latent variables explicitly captures both the concept and the counterfactual latent space, thus promoting structural interpretability, unlike existing structured latent variable models.

**Relation with counterfactual models.** Existing DL methods generating counterfactuals typically offer post-hoc explanations (Ghandeharioun et al., 2021; Abid et al., 2022) or require interpretable input features (Guyomard et al., 2023; Pawelczyk et al., 2020; Mahajan et al., 2020). However, applying post-hoc methods to unstructured data (e.g., pixels instead of concepts) significantly reduces counterfactuals' interpretability (Kim et al., 2018). In contrast, CF-CBMs: (i) are data-agnostic, extracting interpretable counterfactuals regardless of data type, (ii) provide answers to "how?" questions by allowing interventions, (iii) modify the decision-making process by optimizing counterfactuals through joint optimization leading to relying on fewer concepts and (iv) support both concept and task-driven interventions (Table 4). As a side effect, concept-based counterfactuals may also introduce benefits in terms of privacy as sensitive and unique details embedded in input data (e.g., the image of a radiography) remain concealed, thus mitigating the risk of data extraction through repeated model queries (Goethals et al., 2023; Naretto et al., 2022; Pawelczyk et al., 2022).

**Broader Impact.** The ability to answer all the three questions represents a fundamental step to calibrate human trust and enhance human-machine interactions. In this work, we show how combining existing interpretable models (such as CBMs) and post-hoc counterfactual generators may represent a possible solution to address "what?", "how?", and "why not?" queries. Experimental results show that training the counterfactual generator jointly with the CBM leads to two key improvements: (i) it alters the model's decision-making process, making the model rely on fewer important concepts (leading to simpler explanations), and (ii) it significantly increases the causal effect of concept interventions on class predictions, making the model more responsive to these changes. Besides, the ability to perform task-driven interventions may provide actionable insights to improve CF-CBMs' predictions whenever users observe a mispredicted class label but do not know how to intervene on concepts. Finally, the proposed latent variable approach enables to efficiently model concept dependencies while generating multiple counterfactuals for a given input. This represents a key advantage w.r.t. iterative methods in practical applications where proposing a variety of interpretable counterfactuals on the fly represents a fundamental advantage.

## ACKNOWLEDGMENTS

GD acknowledges support from the European Union's Horizon Europe project SmartCHANGE (No. 101080965). PB acknowledges support from Swiss National Science Foundation projects TRUST-ME (No. 205121L_214991) and IMAGINE (No. TMPFP2_224226). FG has been supported by the Partnership Extended PE00000013 - "FAIR - Future Artificial Intelligence Research" - Spoke 1 "Human-centered AI". MG acknowledges support from Swiss National Science Foundation projects XAI-PAC (No. PZ00P2_216405). GM acknowledges support from the KU Leuven Research Fund (STG/22/021, CELSA/24/008) and from the Flemish Government under the "Onderzoeksprogramma Artificiële Intelligentie (AI) Vlaanderen" programme. This work was also supported by the EU Framework Program for Research and Innovation Horizon under the Grant Agreement No 101073307 (MSCA-DN LeMuR).

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

## A  Ammortized inference and ELBO Derivation

To derive the ELBO objective function defined in Section 3.1, we start from the maximization of the log-likelihood of the tuple $(c, y, y')$:

$$\log p(c, y, y') = \log \int_{c', z, z'} p(c, c', y, y', z, z') dc' dz dz'$$

where $p(c, c', y, y', z, z')$ factorizes as in Section 3.1. We consider the variational approximation $q(z, z', c'|x, y, c, y') = q(z|x)q(z'|z, c, y, y')p(c'|z')$ thus exploiting the observation of $x$ to better infer $z$:

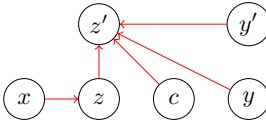

We have:

$$\log p(c, y, y') = \log \int \frac{q(z|x)q(z'|z, c, y, y')p(c'|z')}{q(z|x)q(z'|z, c, y, y')p(c'|z')} \cdot$$
$$\cdot p(c, c', y, y', z, z') dc' dz dz'$$

and, given the Jensen's inequality, we obtain:

$$\log p(c, y, y') \geq \int q(z|x)q(z'|z, c, y, y')p(c'|z') \cdot$$
$$\cdot \log \frac{p(c, c', y, y', z, z')}{q(z|x)q(z'|z, c, y, y')p(c'|z')} dc' dz dz'$$

that can be rewritten as:

$$\log p(c, y, y') \geq \mathbb{E}_z[\log(c|z)] +$$
$$\log p(y|c) +$$
$$\mathbb{E}_{z, z', c'}[\log p(y'|c')] -$$
$$D_{KL}[p(z)|q(z|x)] -$$
$$D_{KL}[p(z')|q(z'|z, c, y, y')].$$

## B  Take the best bet or contemplate a multiverse?

All these functionalities can be used in two ways: the "best bet" mode and "multiverse" mode. In the "best bet" mode, CF-CBMs provide maximum a posteriori estimates by taking the mode of posterior distributions. Using this mode, users will be provided with the most probable counterfactual for each input. In "multiverse" mode instead, CF-CBMs sample from posterior distributions. This allows the model to generate multiple counterfactuals for each input, thus allowing users to choose among a variety of actions/interventions. For instance, in a healthcare scenario, the "best bet" mode might suggest the most probable alternative treatment plan based on patient data, while the "multiverse" mode could present a range of treatment strategies under varying clinical conditions, each representing a different plausible future.

## C  Experimental details

### C.1  Data & task setup

In our experiments we use five different datasets commonly used to evaluate CBMs:

- **dSprites** (Matthey et al., 2017) — It is composed of images of one of three objects (square, ellipse, heart) in different positions and of different sizes. Starting from it, we design a binary task, which is to predict if in the image there is at least a square or a heart. We also combine initial images to obtain new ones with more than one object. The concepts for this task are: (1) the presence of a square, (2) the presence of an ellipse, (3) the presence of a heart, (4) the presence of two objects, (5) objects are red, (6) objects are green, (7) objects are blue.

- **MNIST addition** — The task of this dataset is to predict the sum of two MNIST digits (Deng, 2012). The concepts of this task are the one-hot encoding of the first and the second digits concatenated together.

- **CUB** (Welinder et al., 2010) — It contains pictures of birds and the final task is to predict their species (among 200 species). The concepts are 118 bird features that human annotators selected.

- **CIFAR10**(Krizhevsky et al.) — This dataset contains images of objects, with the task of classifying them into one of 10 classes. Concepts are automatically extracted using the method proposed in Oikarinen et al. (2023), resulting in 143 concepts.

- **SIIM Pneumothorax**(Zawacki et al., 2019) — This dataset contains X-ray scans of lungs, with the task of determining whether the patient has pneumothorax. Concepts are automatically extracted following Oikarinen et al. (2023), leveraging CXR-CLIP, resulting in 19 concepts.

## C.2 EVALUATION METRICS

In our analysis we use the following metrics. **What? (prediction generalization)**: we compute the Area Under the Receiver Operating Characteristic Curve (Hand & Till, 2001) for concepts and tasks (*ROC AUC* ($\uparrow$). **How? (intervention impact)**: we compute the impact of each concept on the downstream task, inspecting the task predictor $f$ (Yuksekgonul et al., 2022). For each class, we look at the related weights of the task predictor (a Linear layer), one weight for each concept. In this way, it is visible the impact of each concept on each specific class. Then, we evaluate the impact of intervening on confounding concepts for class predictions (Koh et al., 2020). We modify the dSprites task to highly correlate the colors with the shapes and therefore the task label. For instance, in 85% of samples with a positive label (presence of at least one square or heart), the objects are green, while in 85% of the samples with a negative label, the objects are red or blue. **Why not? (counterfactual actionability)**: drawing from previous works on counterfactuals, we compute: (i) the **validity**($\uparrow$) (Wachter et al., 2017) by checking whether the model predicts the desired class labels based on the generated counterfactual; (ii) the **proximity** ($\downarrow$) (Pawelczyk et al., 2020) which evaluates counterfactuals' "reliability" as their similarity w.r.t. training samples; (iii) the **time** ($\downarrow$) (Romashov et al., 2022) to generate a counterfactual for a set of samples.

In addition, we propose the following metrics to assess the quality of counterfactuals: (i) the **$\Delta$-Sparsity** ($\downarrow$) which evaluates user's "wasted efforts" by counting the number concepts changed (Guo et al., 2023) w.r.t. the minimal number of changes that would have generated a counterfactual according to the dataset:

$$\Delta Sparsity = |OptimalSparsity - Sparsity|$$

where $OptimalSparsity$ represents the mean Hamming distance between each concept vector in the test set and the closest concept vector of a drawn random class $y'$, found using a "brute-force" approach and $Sparsity$ represents the mean Hamming distance between the predicted concepts $\hat{c}$ and the predicted concept counterfactuals $\hat{c}'$; (ii) the "plausibility" (Wachter et al., 2017) as the **Intersection over Union** ($IoU$ ($\uparrow$)) (Jaccard, 1912) between counterfactuals and ground-truth concept vectors; (iii) the **variability** ($\uparrow$) which evaluates counterfactuals' "diversity", as the cardinality ratio between the set of counterfactuals generated and the set of training concept vectors

$$Variability = \frac{|c'|}{|c|}$$

where $|c'|$ represents the cardinality of the set composed by all $\hat{c}'$, while $|c|$ represents the cardinality of the set composed by all $c$; (iv) finally, we measure the **concept accuracy of generated interventions**

Table 5: Model Hyperparameters shared across all models. During the training, we select the best checkpoint for each model.

| Hyperparams | dSprites | MNIST add | CUB | CIFAR10 | SIIM Pneumothorax |
|---|---|---|---|---|---|
| Epochs | 75 | 150 | 150 | 30 | 100 |
| Learning rate | 0.005 | 0.005 | 0.005 | 0.005 | 0.005 |
| Hidden size | 128 | 128 | 128 | 128 | 128 |
| Batch size | 1024 | 1024 | 1024 | 1024 | 1024 |

(*Acc Int.* ($\uparrow$)) w.r.t. ground-truth optimal interventions that evaluate the model's ability to generate potential concept interventions. Following Espinosa Zarlenga et al. (2022), we inject noise on predicted concepts $\hat{c}$ to reduce task accuracy (randomly flip the value of some concepts), then we sample counterfactuals, conditioning on the ground-truth label $y$, and use them to fix mispredicted concept labels automatically.

$$AccInt. = \frac{\sum_{i \in c} 1_{c_i = \hat{c}'_i}}{|c|}$$

where $c_i$ represents the concept ground truth for the sample $i$, while $\hat{c}'_i$ represents the counterfactual generated for the sample $i$ that tries to recover $c$.

## C.3 BASELINES AND IMPLEMENTATION DETAILS

**Hyperparameters** To train our models and the baselines we selected the best hyperparameters according to our experiments. Table 5 shows the number of epochs, learning rate, and embedding size in the latent space, batch size for each dataset. They are shared among the baselines, and we took the best checkpoint out of the entire training for each model. In addition, Table 6 illustrates the parameters used to weight each term in the loss for all the methods. These hyperparameters were chosen through validation on a subset of the training to achieve the best possible trade-off across all metrics important for counterfactual evaluation, such as validity, proximity, and sparsity, for both our method and the baselines. While our method introduces more hyperparameters, it provides greater flexibility to specify the desired objective. As counterfactual metrics often conflict, creating inherent trade-offs (as shown in our experiments), there is no one-size-fits-all solution. Thus, optimizing the counterfactual generator based on the metric of interest is crucial, and our method facilitates this. Each hyperparameter in the loss has a specific role in steering the model toward a particular behavior:

- $\lambda_1$ (task-related) and $\lambda_2$ (concept-related) are important in all configurations as it remains the main goal of the model.

- $\lambda_3$ prioritizes validity, encouraging counterfactuals with predictions matching $y\prime$, though potentially compromising proximity or sparsity.

- $\lambda_4$ and $\lambda_5$ emphasize generating more realistic counterfactuals, potentially reducing proximity, which may trade off against validity and sparsity.

- $\lambda_6$ and $\lambda_7$ promotes sparsity by reducing the number of changes from the input, which may trade off against validity. However, we observed that $\lambda_6$ has minimal influence on the optimization process, with $\lambda_7$ being the key hyperparameter driving this behaviour.

This flexibility makes our model more versatile and practical compared to baselines, which have limited or partial support for these trade-offs (e.g., VAECF has the possibility to be flexible on validity). To show these behaviours we perform an ablation study on MNIST add that is shown in Table 7.

**Counterfactual CBM implementation details** To improve the training process of our models we decided to relax some assumptions we made in Section 3.1. Employing a Bernoulli distribution to predict concepts and a Categorical distribution for the class prediction would make the training process more difficult, ruining the gradient in the backpropagation process. Therefore, we predict them directly with the output of the concept predictor and the task predictor applying a Sigmoid function and a Softamx on top of them, respectively. To optimise them, we choose to use a Binary Cross Entropy loss for the concept loss, and a Cross Entropy Loss for the task loss. All this is

Table 6: Loss weights

| Method | | Dataset | | | | |
|---|---|---|---|---|---|---|
| | Loss Term | dSprites | MNIST add | CUB | CIFAR10 | SIIM Pneumothorax |
| CBM | Concept | 1.0 | 1.0 | 1.0 | 1.0 | 1.0 |
| | Task | 0.1 | 0.1 | 0.1 | 0.1 | 0.1 |
| CCHVAE | Reconstruction | 1.0 | 1.0 | 1.0 | 1.0 | 1.0 |
| | KL | 0.5 | 0.5 | 0.5 | 0.5 | 0.5 |
| VAECF | Reconstruction | 3.0 | 1.3 | 1.0 | 1.3 | 1.3 |
| | KL | 1.0 | 1.0 | 2.0 | 1.0 | 1.0 |
| | Validity | 15.0 | 15.0 | 15.0 | 20.0 | 20.0 |
| VCNET | Concept | 1.0 | 1.0 | 1.0 | 1.0 | 2.0 |
| | Task | 0.5 | 0.5 | 0.5 | 0.5 | 0.5 |
| | Reconstruction | 0.5 | 0.5 | 0.5 | | 0.2 |
| | KL | 0.8 | 0.8 | 0.8 | 0.8 | 0.8 |
| CF-CBM | Concept | 10.0 | 10.0 | 1.0 | 10.0 | 10.0 |
| | Task | 0.7 | 1.0 | 0.1 | 1.0 | 1.0 |
| | Validity | 0.3 | 0.2 | 0.02 | 0.2 | 0.2 |
| | $z$ KL | 1.2 | 2.0 | 0.2 | 2.0 | 2.0 |
| | $z'$ KL | 1.2 | 2.0 | 0.2 | 2.0 | 2.0 |
| | Prior distance | 1.0 | 1.7 | 0.2 | 1.7 | 1.7 |
| | Posterior distance | 0.6 | 0.55 | 0.03 | 0.4 | 0.4 |

Table 7: Performance comparison for different objective focuses with varying $\lambda$ values on the MNIST add dataset.

| Objective Focus | $\lambda_3$ | $\lambda_4$ & $\lambda_5$ | $\lambda_7$ | Validity (%) | Sparsity (%) | Proximity (%) |
|---|---|---|---|---|---|---|
| Higher focus on validity | 0.4 | 2 | 0.55 | $99.1 \pm 0.2$ | $19.0 \pm 0.4$ | $4.3 \pm 0.7$ |
| Higher focus on proximity | 0.2 | 4 | 0.55 | $93.6 \pm 1.6$ | $14.2 \pm 0.4$ | $3.6 \pm 0.0$ |
| Higher focus on sparsity | 0.2 | 2 | 0.70 | $74.4 \pm 0.01$ | $6.1 \pm 0.4$ | $2.6 \pm 0.0$ |

done in a jointly training to fully leverage the advantages outlined in Section 5.2. This means that all component are trained together. However, it is also possible to train the model in a post-hoc manner—first training the concept encoder and predictor, followed by the counterfactual generator, or performing a warm-up phase for the encoder and predictor to better initialize their weights before continuing with joint training.

**Code, licenses and hardware** For our experiments, we implement all baselines and methods in Python 3.9 and relied upon open-source libraries such as PyTorch 2.0 (Paszke et al., 2019) (BSD license), PytorchLightning v2.1.2 (Apache Licence 2.0), Sklearn 1.2 (Pedregosa et al., 2011) (BSD license). In addition, we used Matplotlib (Hunter, 2007) 3.7 (BSD license) to produce the plots shown in this paper. The datasets we used are freely available on the web with licenses: dSprites (Apache 2.0) MNIST (CC 3.0 DEED) and CUB (MIT License). We will publicly release the code with all the details used to reproduce all the experiments under an MIT license. The experiments were performed on a device equipped with an M3 Max and 36GB of RAM, without the use of a GPU. Approximately 80 hours of computational time were utilized from the start of the project, whereas reproducing the experiments detailed here requires only 10 hours.

# D  FURTHER METRICS AND RESULTS

In addition to the result present in Section 5, Table 8 shows the task and concept performance of our model and the baselines. Moreover, Figure 6 illustrates the Acc Int. achieved by all models with different levels of noise injected at the concept level. It is visible how CF-CBM achieves stable Acc Int. results with different level of noise injected at the concept level. This is not the case for the other baselines. Figure 7 shows the distribution of the CaCE score obtained in MNIST add and CUB in the post-hoc and joint fashions. It allows to directly see the distributions used to compute the metrics in Table 2. Finally, Figure 8 shows the time efficency of all the method in generating counterfactuals. Generation efficiency is a key feature for real-time applications. In this respect, CF-CBMs significantly outperform iterative models such as C-CHVAE and BayCon. These models quickly struggle in large search spaces involving a high number of concepts: generating a

Table 8: **CF-CBM attains generalization performance of standard CBMs** on concepts and class labels.

| | dsprites | | MNIST add | | CUB | | CIFAR10 | | SIIM Pneumothorax | |
|---|---|---|---|---|---|---|---|---|---|---|
| | Task (%) (↑) | Concept (%) (↑) | Task (%) (↑) | Concept (%) (↑) | Task (%) (↑) | Concept (%) (↑) | Task (%) (↑) | Concept (%) (↑) | Task (%) (↑) | Concept (%) (↑) |
| Black Box | 99.9 ± 0.0 | - | 98.4 ± 0.0 | - | 94.4 ± 0.0 | - | 99.7 ± 0.0 | - | 95.1 ± 0.2 | |
| CBM | 99.6 ± 0.1 | 98.8 ± 0.1 | 98.4 ± 0.0 | 99.7 ± 0.0 | 93.0 ± 0.0 | 85.3 ± 0.1 | 99.3 ± 0.0 | 99.1 ± 0.0 | 89.4 ± 0.6 | 99.6 ± 0.1 |
| CEM | 99.8 ± 0.1 | 98.6 ± 0.1 | 98.4 ± 0.0 | 99.5 ± 0.0 | 93.3 ± 0.1 | 85.7 ± 0.1 | 99.8 ± 0.0 | 99.2 ± 0.0 | 94.2 ± 0.1 | 99.5 ± 0.0 |
| BayCon | 99.6 ± 0.0 | 98.7 ± 0.1 | 98.4 ± 0.0 | 99.7 ± 0.0 | 93.1 ± 0.1 | 85.3 ± 0.1 | 99.4 ± 0.0 | 99.1 ± 0.0 | 89.3 ± 0.8 | 99.6 ± 0.1 |
| CCHVAE | 99.5 ± 0.1 | 98.9 ± 0.1 | 98.4 ± 0.0 | 99.7 ± 0.0 | 93.2 ± 0.1 | 85.4 ± 0.1 | 99.3 ± 0.0 | 99.2 ± 0.0 | 89.6 ± 0.5 | 99.3 ± 0.1 |
| VAECF | 99.5 ± 0.0 | 98.8 ± 0.1 | 98.4 ± 0.0 | 99.7 ± 0.0 | 92.9 ± 0.2 | 85.4 ± 0.1 | 99.3 ± 0.0 | 99.2 ± 0.0 | 89.5 ± 0.8 | 99.6 ± 0.1 |
| VCNET | 99.7 ± 0.0 | 98.6 ± 0.1 | 98.2 ± 0.0 | 99.6 ± 0.0 | 89.6 ± 0.5 | 80.7 ± 0.1 | 99.1 ± 0.1 | 97.4 ± 0.2 | 91.4 ± 0.7 | 99.2 ± 0.0 |
| CF-CBM | 99.7 ± 0.0 | 98.7 ± 0.1 | 97.0 ± 0.1 | 99.3 ± 0.0 | 91.4 ± 0.2 | 84.8 ± 0.2 | 99.4 ± 0.0 | 96.6 ± 0.0 | 92.0 ± 0.5 | 98.2 ± 0.0 |

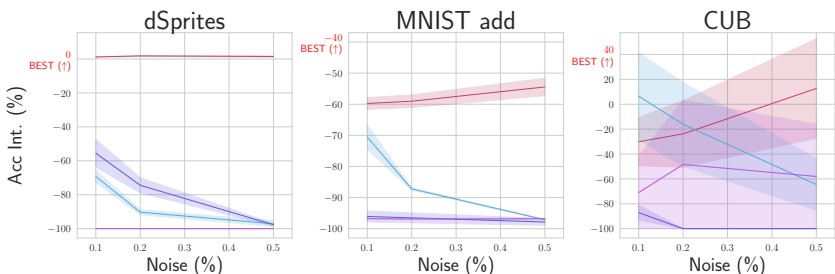

Figure 6: **CF-CBMs attain similar results while applying different levels of noise** on predicted concept labels.

counterfactual takes more than a minute on dSprites (7 concepts) and up to a few hours on CUB (118 concepts). CF-CBMs instead scale well with the size of the data set by requiring less than a second on dSprites and a few seconds on CUB, on par with VAECF and VCNet. This efficiency enables a wider range of applications requiring a high number of concepts and quick feedbacks.

## E  COUNTERFACTUALS

Tables 9 and 10 present the top-3 most common counterfactual examples for each method on dSprites, MNIST addition, and SIIM Pneumothorax, respectively. The tables display only the active concepts to maintain clarity, as including all concepts would make them difficult to interpret. All other concepts are considered inactive. Examining these three tables—especially the first two, which are easier to understand without requiring domain-specific knowledge—it is clear that the counterfactuals generated by CF-CBM are the most reasonable. In Table 9, CF-CBM consistently generates feasible states by modifying the minimum number of concepts. In contrast, VCNET fails to activate any color concepts, leading to suboptimal changes and unfeasible states. Similarly, CCHVAE activates the "Two obj" concept, which is irrelevant and fails to influence the prediction, effectively wasting a change. For MNIST addition, CF-CBM once again generates valid counterfactuals, while other baselines occasionally produce counterfactuals with zero, one, or three active concepts—an impossible scenario, as each sample must always contain two digits. This issue is particularly prominent in VCNET and VAECF, which rely heavily on the fuzziness of concept values for decision-making, allowing scenarios where no active concepts still predict different labels. Lastly, we include a single example of counterfactuals per model for CUB and CIFAR10 for completeness. Due to the large number of concepts in these datasets, presenting multiple examples in an organized format would be challenging.

**CUB**:

- Baycon:
    - Factual: Bill Shape - Hooked (Seabird), Wing Color - Grey, Underparts Color - White, Breast Pattern - Solid, Breast Color - White, Throat Color - White, Eye Color - Black, Bill Length - Same as Head, Nape Color - White, Belly Color - White, Size - Medium (9-16 in), Back Pattern - Solid, Tail Pattern - Solid, Belly Pattern - Solid, Wing Pattern - Solid - Task: Brandt Cormorant
    - Counterfactual: CF: Bill Shape - Hooked (Seabird), Wing Color - Grey, Upperparts Color - Black, Underparts Color - Black, Breast Pattern - Solid, Breast Pattern - Striped, Upper Tail Color - White, Head Pattern - Plain, Breast Color - Black, Throat Color -

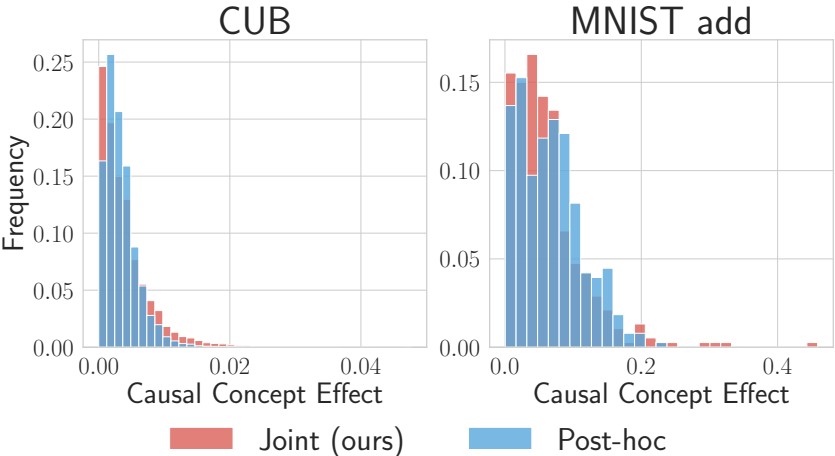

Figure 7: The distribution of the CaCE score of the CUB and MNIST add dataset obtained by the end-to-end (Joint) approach and by the post-hoc one.

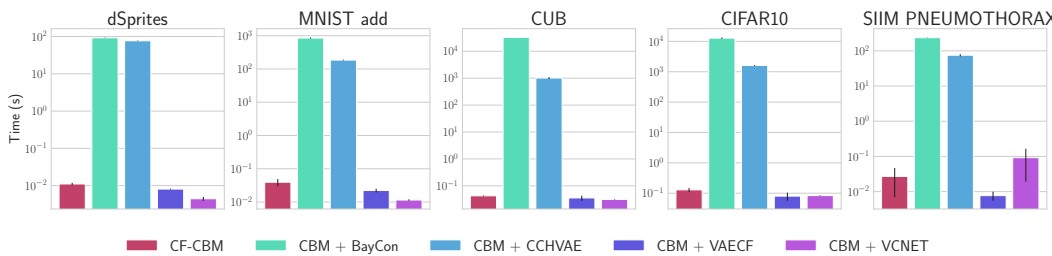

Figure 8: **CF-CBMs efficiently generate counterfactuals on the fly**, on par with the fastest generators.

Yellow, Eye Color - Black, Bill Length - Same as Head, Forehead Color - White, Under Tail Color - Black, Nape Color - Yellow, Belly Color - Grey, Belly Color - Black, Size - Medium (9-16 in), Back Pattern - Solid, Tail Pattern - Solid, Belly Pattern - Solid, Leg Color - Black, Wing Pattern - Solid - Task: Black Footed Albatross

- CCHVAE:
  - Factual: Bill Shape - Cone, Wing Color - Brown, Upperparts Color - Brown, Upperparts Color - Buff, Underparts Color - White, Breast Pattern - Striped, Back Color - Brown, Tail Shape - Notched, Upper Tail Color - Brown, Upper Tail Color - Buff, Breast Color - White, Throat Color - White, Eye Color - Black, Bill Length - Shorter than Head, Forehead Color - Brown, Belly Color - White, Wing Shape - Rounded, Size - Small (5-9 in), Body Shape - Perching-like, Back Pattern - Striped, Primary Color - Brown, Crown Color - Brown, Wing Pattern - Striped, - Task: Purple Finch
  - Counterfactual: Bill Shape - Hooked, Wing Color - Black, Upperparts Color - Black, Underparts Color - Black, Breast Pattern - Solid, Back Color - Black, Upper Tail Color - Black, Head Pattern - Plain, Breast Color - Black, Throat Color - Black, Eye Color - Black, Bill Length - Shorter than Head, Forehead Color - Black, Under Tail Color - Black, Nape Color - Black, Belly Color - Black, Wing Shape - Rounded, Size - Small (5-9 in), Body Shape - Perching-like, Back Pattern - Solid, Tail Pattern - Solid, Belly Pattern - Solid, Primary Color - Black, Leg Color - Black, Bill Color - Black, Crown Color - Black, Wing Pattern - Solid, Task: Shiny Cowbird

- VAECF:
  - Factual: Breast Pattern - Solid, Eye Color - Black, Bill Length - Shorter than Head, Size - Small (5-9 in), Size - Very Small (3-5 in), Back Pattern - Solid, Tail Pattern - Solid, Belly Pattern - Solid, Leg Color - Black, Bill Color - Black, Wing Pattern - Multi-Colored - Task: Rufous Hummingbird

- – Counterfactual: Wing Color - Black, Upperparts Color - Black, Underparts Color - Black, Breast Pattern - Solid, Back Color - Black, Upper Tail Color - Black, Head Pattern - Plain, Breast Color - Black, Throat Color - Black, Eye Color - Black, Bill Length - Same as Head, Forehead Color - Black, Under Tail Color - Black, Nape Color - Black, Belly Color - Black, Back Pattern - Solid, Tail Pattern - Solid, Belly Pattern - Solid, Primary Color - Black, Leg Color - Black, Bill Color - Black, Crown Color - Black, Wing Pattern - Solid - Task: American Crow
- VCNET:
  - – Factual: Underparts Color - White, Breast Pattern - Solid, Eye Color - Black, Bill Length - Shorter than Head, Belly Color - White, Size - Small (5-9 in), Body Shape - Perching-like, Back Pattern - Solid, Belly Pattern - Solid - Task: White Pelican
  - – Counterfactual: Bill Shape - Dagger, Wing Color - Grey, Wing Color - White, Upperparts Color - Grey, Upperparts Color - White, Underparts Color - White, Breast Pattern - Solid, Back Color - Grey, Back Color - White, Upper Tail Color - White, Head Pattern - Plain, Breast Color - White, Throat Color - White, Eye Color - Black, Bill Length - Same as Head, Forehead Color - White, Under Tail Color - White, Nape Color - White, Belly Color - White, Size - Medium (9-16 in), Back Pattern - Solid, Back Pattern - Multi-Colored, Tail Pattern - Solid, Belly Pattern - Solid, Primary Color - Grey, Primary Color - White, Crown Color - White, Wing Pattern - Solid - Task: Ivory Gull
- CF-CBM:
  - – Factual: Bill Shape - Cone, Wing Color - Brown, Wing Color - Buff, Upperparts Color - Brown, Upperparts Color - Black, Upperparts Color - Buff, Underparts Color - White, Breast Pattern - Striped, Back Color - Brown, Back Color - Buff, Tail Shape - Notched, Upper Tail Color - Brown, Upper Tail Color - Buff, Breast Color - Buff, Throat Color - Buff, Eye Color - Black, Bill Length - Shorter than Head, Under Tail Color - Buff, Wing Shape - Rounded, Size - Small (5-9 in), Body Shape - Perching-like, Back Pattern - Striped, Belly Pattern - Solid, Primary Color - Brown, Bill Color - Buff, Wing Pattern - Striped - Task: Harris Sparrow
  - – Counterfactual: Bill Shape - Dagger, Wing Color - Grey, Wing Color - White, Upperparts Color - Grey, Upperparts Color - White, Underparts Color - White, Breast Pattern - Solid, Back Color - Grey, Back Color - White, Upper Tail Color - White, Head Pattern - Plain, Breast Color - White, Throat Color - White, Eye Color - Black, Bill Length - Same as Head, Forehead Color - White, Under Tail Color - White, Nape Color - White, Belly Color - White, Size - Medium (9-16 in), Back Pattern - Solid, Belly Pattern - Solid, Primary Color - Grey, Primary Color - White, Crown Color - White - Task: Western Gull

**CIFAR10**:

- Baycon:
  - – Factual: Hunter, beak, bed, birdfeeder, birdhouse, bit, branch, cage, captain, cat bed, cat food dish, cat toy, collar, crew, dashboard, deck, dog bowl, fly, food bowl, forest, four-legged mammal, gear shift, green or brown body, grille, hitch, lead rope, leaf, litter box, long neck, mane and tail, mast, mosquito, nest, net, nose, pedal, person, pointed front end, pond, port, reddish-brown coat, rifle, scratching post, seat, seatbelt, short, stocky build, small, lithe body, spider, steering wheel, tail, tire, tough, durable frame, tree, wet nose, wheel, wide mouth, worm, an engine, animal, cargo, engines, engines on the wings, feathers, food, four legs, four round, black tires, fur, grass, hooves, insects, large sails or engines, large, brown eyes, large, bulging eyes, lily pads, living thing, long ears, machine, mammal, multiple decks, multiple sails, nature, object, organism, passengers, pointed ears, quadruped, side windows, the ocean, trees, two headlights, vertebrate, vessel, webbed feet, whiskers, white spots on the coat, wings, woods - Task: Cat
  - – Counterfactual: Hunter, beak, bed, beetle, bird feeder, birdfeeder, birdhouse, bit, boat, branch, captain, cat food dish, cat toy, collar, copilot, crew, dashboard, deck, dog bowl, field, flat back end, flat front and back, flight attendant, fly, food bowl, gear

shift, grille, hitch, large size, leaf, leash, lily pad, litter box, long neck, mast, mosquito, net, nose, passenger, pedal, person, pilot, pointed front end, port, rider, rifle, road, rudder, runway, saddle, seat, seatbelt, spider, steering wheel, suitcase, tail, tire, toy, tree, wet nose, wheel, wide mouth, windshield, worm, an engine, animal, antlers, cargo, engines, engines on the wings, feathers, food, four round, black tires, fur, grass, hooves, insects, landing gear, large sails or engines, large, brown eyes, large, bulging eyes, lily pads, living thing, machine, mammal, multiple decks, multiple sails, object, organism, passengers, pointed ears, quadruped, side windows, the ability to fly, the ocean, transportation, trees, two headlights, vertebrate, vessel, watercraft, webbed feet, whiskers, white spots on the coat, wings - Task: Airplane

- CCHVAE:
  - Factual: boat, captain, deck, dock, port, large sails or engines, multiple decks, multiple sails, the ocean, vessel, watercraft - Task: Ship
  - Counterfactual: None - Task: Horse

- VAECF:
  - Factual: beak, bed, bed for carrying cargo, beetle, bird feeder, birdfeeder, birdhouse, bit, boat, branch, cage, captain, cat bed, cat toy, collar, copilot, crew, dashboard, deck, dock, dog bowl, field, flat back end, flat front and back, flight attendant, fly, gear shift, grille, halter, hitch, large body, large, metal body, lead rope, leash, litter box, long neck, mast, mosquito, net, pedal, person, pilot, pointed front end, port, rifle, road, rudder, runway, saddle, scratching post, seat, seatbelt, short, stocky build, small, lithe body, spider, steering wheel, suitcase, tail, tire, tough, durable frame, toy, tree, wheel, windshield, worm, an engine, cargo, engines, engines on the wings, four legs, four round, black tires, four wheels, fur, grass, hooves, insects, landing gear, large sails or engines, large, bulging eyes, long hind legs for jumping, long, thin legs, machine, mammal, multiple decks, multiple sails, object, organism, passengers, quadruped, the ability to fly, the ocean, transportation, trees, vertebrate, vessel, watercraft, webbed feet, wings - Task: Airplane
  - Counterfactual: runway, the ability to fly - Task: Horse

- VCNET:
  - Factual: barn, bed, bed for carrying cargo, beetle, cab for the driver, copilot, dashboard, deck, driver, flat back end, flat front and back, gear shift, green or brown body, grille, hitch, large body, large, metal body, large, rectangular body, passenger, pointed front end, road, seatbelt, short, stocky build, steering wheel, suitcase, tire, tough, durable frame, trailer, wheel, windshield, an engine, cargo, engines, four round, black tires, four wheels, multiple decks, passengers, side windows, taillights, transportation, two headlights, vessel - Task: Cat
  - Counterfactual: bed for carrying cargo, boat, captain, crew, deck, dock, flat front and back, mast, port, rudder, an engine, cargo, engines, large sails or engines, multiple decks, multiple sails, passengers, side windows, the ocean, transportation, vessel, watercraft - Task: Ship

- CF-CBM:
  - Factual: bed for carrying cargo, boat, captain, deck, dock, mast, port, rudder, an engine, engines, large sails or engines, multiple decks, multiple sails, passengers, side windows, the ocean, transportation, vessel, watercraft - Task: Ship
  - Counterfactual: barn, bridle, field, four-legged mammal, halter, lead rope, leash, mane and tail, pasture, reins, rider, saddle, animal, four legs, hooves, long ears, long hind legs for jumping, long, thin legs, quadruped, white spots on the coat - Task: Horse

Table 9: Comparison of the most three common counterfactuals for different models on the dSprites dataset. We are showing just the active concepts.

| Model | Factual | | Counterfactuals | |
|---|---|---|---|---|
| | Concepts | Task | Concepts | Task |
| BayCon | Heart Green | 1 | Ellipse Heart Green | 0 |
| | Square Green | 1 | Ellipse Heart Green | 0 |
| | Ellipse Blue | 0 | Square Ellipse Blue | 1 |
| CCHVAE | Heart Green | 1 | Ellipse Red | 0 |
| | Heart Green | 1 | Ellipse Green | 0 |
| | Heart Green | 1 | Ellipse Two obj Green | 0 |
| VAECF | Heart Green | 1 | Ellipse | 0 |
| | Heart Green | 1 | Ellipse Red | 0 |
| | Square Blue | 1 | Ellipse Blue | 0 |
| VCNET | Heart Green | 1 | Ellipse | 0 |
| | Heart Red | 1 | Ellipse | 0 |
| | Ellipse Red | 0 | Square Heart | 1 |
| CF-CBM | Heart Green | 1 | Ellipse Green | 0 |
| | Heart Red | 1 | Ellipse Red | 0 |
| | Square Heart Two obj Green | 1 | Ellipse Two obj Green | 0 |

Table 10: Comparison of the most three common counterfactuals for different models on the MNIST add dataset. We are showing just the active concepts.

| Model | Factual | | Counterfactuals | |
|---|---|---|---|---|
| | Concepts | Task | Concepts | Task |
| BayCon | 5 | 10 | 7, 4 | 0 |
| | 2, 0 | 2 | 0, 4, 0 | 0 |
| | 8, 7 | 15 | 4, 0, 7 | 0 |
| CCHVAE | 1, 2 | 3 | 0, 0 | 0 |
| | 2, 9 | 11 | 9, 9 | 18 |
| | 1, 1 | 2 | 9 | 17 |
| VAECF | 1, 1 | 2 | 8 | 16 |
| | 7, 3 | 10 | 1, 1 | 2 |
| | 2, 1 | 3 | None | 10 |
| VCNET | 2, 9 | 10 | None | 12 |
| | 0, 6 | 7 | 9, 9 | 18 |
| | 3, 8 | 7 | None | 15 |
| CF-CBM | 6, 6 | 12 | 9, 9 | 18 |
| | 5, 0 | 5 | 7, 8 | 15 |
| | 0, 1 | 1 | 0, 2 | 2 |

Table 11: Comparison of the most three common counterfactuals for different models on the X-ray dataset. We are showing just the active concepts.

| Model | Factual | | Counterfactuals | |
|---|---|---|---|---|
| | Concepts | Task | Concepts | Task |
| BayCon | Emphysematous changes, Hyperinflated lung | 1 | Diaphragmatic contour, Pleural effusion | 0 |
| | Collapsed lung, Mediastinal shift | 1 | Collapsed lung, Mediastinal shift, Lung consolidation, Air distribution pattern | 0 |
| | Subcutaneous emphysema, Pleural effusion, Emphysematous changes | 1 | Diaphragmatic contour, Rib fractures, Emphysematous changes, Cardiac shadow abnormality, Deep sulcus sign, Radiodensity changes | 0 |
| CCHVAE | Visible pleural line, Rib fractures, Absent lung markings, Lung fibrosis, Mediastinal contour Cardiac shadow abnormality, Deep sulcus sign, Symmetry analysis, Air distribution pattern | 0 | Pleural effusion | 1 |
| | None | 0 | None | 1 |
| | Visible pleural line, Diaphragmatic contour, Rib fractures, Pleural effusion, Deep sulcus sign, Hyperinflated lung, Symmetry analysis | 0 | Emphysematous changes, Hyperinflated lung | 1 |
| VAECF | Pleural effusion | 0 | Emphysematous changes, Air distribution pattern | 1 |
| | Subcutaneous emphysema, Emphysematous changes | 1 | Deep sulcus sign | 0 |
| | Collapsed lung, Pleural effusion, Lung consolidation | 0 | Emphysematous changes, Radiodensity changes, Air distribution pattern | 1 |
| VCNET | Pleural effusion | 1 | Emphysematous changes, Hyperinflated lung | 0 |
| | Collapsed lung, Pleural effusion, Lung consolidation | 0 | Subcutaneous emphysema, Emphysematous changes, Hyperinflated lung | 1 |
| | Emphysematous changes, Hyperinflated lung | 1 | Lung consolidation | 0 |
| CF-CBM | Emphysematous changes | 1 | None | 0 |
| | None | 0 | Subcutaneous emphysema, Emphysematous changes | 1 |
| | Absent lung markings, Lung fibrosis, Cardiac shadow abnormality | 0 | Subcutaneous emphysema, Emphysematous changes, Hyperinflated lung | 1 |

