# OpenReview forum: "Counterfactual Concept Bottleneck Models"
_ICLR.cc/2025/Conference — ICLR 2025 Poster_

### Official Review · Reviewer_RbDp · 2024-10-25

**Soundness:** 3
**Presentation:** 3
**Contribution:** 3
**Rating:** 8
**Confidence:** 4

**Summary:**

This work introduces the framework of counter factual CBMs, aimed specifically to increase the variability and depth of inspection question that can be asked towards a CBM. In other words, CF-CBM aims to increase the level of interpretability and interactability of standard CBM models without losing performance.

**Strengths:**

The paper is very well written and structured. It is well motivated and the methodological contribution as far as I can tell is clear and important.

**Weaknesses:**

I just have a few minor remarks which I have added to the questions section.

**Questions:**

- 5.1 I find it important to have the results of this question also in the main paper, e.g., average values, as it is an important one. I would additionally suggest the authors change the wording of the conclusion. It seems that on average CF-CBMs are slightly lowerd in predictive performance. While I don’t think this is a drawback of the proposed emthod, I think stating the method is on par is slightly overstated. That CF-CVM is not necessarily on par is an important information for the reader/user. Again stating this won’t decrease the contribution.

- 5.2. I don’t understand the intuition behind why CF-CBMs should be less prone to confounding factors? What mechanism in their learning is beneficial for this? Maybe an ablation study would be intersting to underscore these findings.

- Please provide code to the paper, e.g., an anonymized github link.

- Lastly, this [1] might be an interesting paper to add to the Introduction as it tackles learning concepts (e.g., for CBMs) focussing on learning inspectable concepts, e.g., via counterfactual querying. Of course [1] focuses on counterfactual concepts, whereas this work focuses on counterfactuals for a model decision.

[1] Wolfgang Stammer, Antonia Wüst, David Steinmann, and Kristian Kersting. “Neural Concept Binder.” In: Advances in Neural Information Processing Systems (NeurIPS) (2024)

---

> ### Author Response · Authors · 2024-11-22
> **Response to Reviewer RbDp**
>
> ### "Rewording the conclusion of 5.1” and add Table 8 in Section 5
> **We rephrased the conclusion of 5.1** as follows:
> “CF-CBMs achieve generalization performance that is close to standard CBMs”.
> Regarding Table 8, we agree with the reviewer that including it in Section 5 would add value. However, the differences between the methods are minimal. Given the limited space and the fact that accuracy is not the primary focus of our work, we prioritized presenting other results instead.
>
> ----
>
> ### Why do CF-CBMs should be less prone to confounding factors?
> **CF-CBMs prioritize the most impactful features for the task**, as demonstrated in Section 5.2. If a set of features has greater predictive power, the model will favor them over the confounder, which has a lower impact. For example, in our experiments, key features are consistently useful for prediction (100% power), whereas the confounder contributes correctly only 85% of the time. This behavior arises from our loss functions (Equation 5), which guide the model to learn decision boundaries perpendicular to meaningful features while minimizing reliance on less informative ones. This happens to facilitate the counterfactual generation process which wants the output to remain close to the input but also in-distribution.
>
> ----
>
> ### “Provide the code”
> **The code is already attached** and can be used to reproduce the results. However, in the final version of the paper we will include the link to a GitHub repository.
>
> ----
>
> ### Additional citation to a paper using counterfactuals in the realm of concept based interpretability
> We thank the reviewer for the suggestion and for sharing this interesting paper with us. **We have added the mentioned reference** in Section 6 (Related Work), as we believe it is more appropriate to discuss it there. As pointed out by the reviewer, [1] is indeed connected to our work, albeit from a different perspective.

---

### Official Review · Reviewer_9mnf · 2024-10-25

**Soundness:** 3
**Presentation:** 4
**Contribution:** 3
**Rating:** 8
**Confidence:** 4

**Summary:**

The paper introduces a concept bottleneck model with the ability to generate counterfactual explanations. The paper aims to answer three questions: what, how and why not. The what questions are answered by showing the concepts used to make predictions. The how questions are answered by changing the concepts and observing how the prediction changes. Finally, why not questions can be answered by providing an alternative outcome and observing how the concepts change to understand what-if situations. The method also enables what the authors call “task-driven” interventions that allow users to correct wrongly predicted concepts. The models are tested on three datasets: dSprites, MNIST add CUB, and tested on various metrics. In general, based on the experiments, the proposed model demonstrates good performance across multiple tasks and datasets.

**Strengths:**

- The paper is easy to read, and the figures are well described and informative.
- Explanations seem to be easy to use.
- The method is tested against many baselines.

**Weaknesses:**

- I am concerned regarding the number of hyperparameters that need to be set for the loss function. It is not clear how important they are, and how much tuning is needed. How easy is it to tune these hyperparameters? How do they affect the results? Is it easier to have separate systems rather than one big as the one proposed?
- The method requires handcrafted concepts, and it might not be easy to come up with these concepts. Further, there might be a lot of labor involved in creating these concepts. Moreover, the predictive performance depends on how well these concepts describe the input features.
- Figure 1: Is it only for illustration or is this actually the result from trained models on x-ray images?
- I might have missed it, but I cannot find the citation for the dataset from which the x-ray images come. If I did not miss it, please cite it so it becomes easier for readers to find it. Moreover, are there any restrictions on showing these images, that is, what is the license of these images?
- I believe the x-ray images to be from the Osteoarthritis Initiative dataset. It is used to motivate the paper, but why is it not used in the experiments? Are there any weaknesses with the method making the dataset unsuitable for experimentation and showcasing results?
- Line 156: Why does z' need to be conditioned on both c and y? Don't we already know y if we know c?
- Line 224: How do we know Equation 5 gives us sparser explanations? Is it because the posterior is moved towards the prior distribution? But this is defined in the latent space. How can we know that the concepts c will be sparse even though the posterior of z is close to the prior?
- How are the models trained? Are certain parts frozen, or are all components trained jointly?
- Is it possible to show readers the explanations themselves for the datasets used and not only Figure 1 for a dataset not used? We have seen many quantitative results, but part of XAI also involves qualitative qualities. How do these explanations perform qualitatively with end-users? Are they useful to human end-users? And who are the intended end-users for these explanations?
- Although we do not need to run post-hoc searches, new black-box components are introduced into the model. How does that affect the overall interpretability of the model? I believe that can also have negative side effects and not only positive influences.

**Questions:**

- Please refer to the weaknesses.

---

> ### Author Response · Authors · 2024-11-22
> **Response to Reviewer 9mnf**
>
> ### Figure 1: Which dataset are the images sourced from? Are there any restrictions on copyright? Why are you not using the Osteoarthritis dataset in experiments?
> **1. The images in the abstract figures come from the Osteoarthritis dataset** present at this link (https://www.kaggle.com/datasets/shashwatwork/knee-osteoarthritis-dataset-with-severity) distributed with license CC4.0. Thanks for pointing out that there is no reference.
>
> **2. We modified all the examples using the SIIM Pneumothorax dataset.**
>
> **3. We added experiments on the SIIM Pneumothorax dataset** (see the common answer for the experiment on this x-ray dataset).
>
> ----
>
> ### Why $z{\prime}$ depends on $c$ and $y$?
> **$y$ is not essential but it avoids unnecessary double learning of the same process**. We agree that $y$ is not strictly necessary, as $c$ already contains the information to predict $y$. However, including $y$ makes this information explicit. Without $y$, the network predicting $z{\prime}$ would also need to learn again what the predictor has already learned, adding unnecessary complexity to the model.
>
> ----
>
> ### Why does Equation 5 ensure sparser explanations?
> **Learning variational latent models with smooth priors (like our VAEs with standard gaussians priors) encourages smoothness in the encoding/decoding scheme, i.e. samples close in the latent space ($z$, $z{\prime}$) tend to be close also in the output space ($c$, $c{\prime}$).** We have added a reference in the paper to [1]. Thanks for pointing this out.
>
> ----
>
> ### How is the model trained?
> **Jointly.**  The model is trained jointly to fully leverage the advantages outlined in Section 5.2. However, it is also possible to train the model in a post-hoc manner—first training the concept encoder and predictor, followed by the counterfactual generator, or performing a warm-up phase for the encoder and predictor to better initialize their weights before continuing with joint training. We have included this detail in Appendix C.3.
>
> ----
>
> ### Are Introduced black-box components affecting interpretability?
> **No, the introduced black-box components do not affect interpretability, just as adding a hidden layer to a CBM does not compromise its interpretability.** Our model remains as interpretable as CBMs, focusing on concept-based interpretability by highlighting the key concepts used for the final prediction. The added complexity is limited to the stages before concept extraction, enabling the encoding of information about possible counterfactuals. Thus, our model retains the interpretability of CBMs while providing the additional capability to explain decisions through counterfactuals. We added this at L158 as follows: “The added complexity introduced with these additional black-box components is limited to the stages before concept extraction, enabling the encoding of information about possible counterfactuals. Thus, our model retains the interpretability of CBMs while providing the additional capability to explain decisions through counterfactuals.”
>
> ----
>
> [1] Kingma, Diederik P. and Max Welling. “Auto-Encoding Variational Bayes.” CoRR abs/1312.6114 (2013): n. pag.

---

> > ### Comment · Reviewer_9mnf · 2024-11-25
> >
> > I would like to thank the authors for thorough responses to both my and the other reviewers' concerns. I will raise my score accordingly.

---

### Official Review · Reviewer_nCvr · 2024-11-03

**Soundness:** 3
**Presentation:** 3
**Contribution:** 3
**Rating:** 6
**Confidence:** 3

**Summary:**

This paper presents a method that combines counterfactual and concept bottleneck that can answer "What?", "How?" and "Why not?" all at once. It leverages latent variable models to generate counterfactuals via variational inference. It achieves comparable classification accuracy to standard CBMs and black-box models and outperforms in generating interpretable, concept-base counterfactuals.

**Strengths:**

1. The method is straightforward and intuitive, combining two existing approaches, counterfactual reasoning and concept bottleneck modeling with moderate novelty. The experiments provide solid evidence of its effectiveness.
2. This paper is well organized, and abstract and introduction effectively set the stage for the paper's major content.
3. The method enhances the interpretability of counterfactuals, making it potentially valuable for many real-life settings like medical applications, as the authors suggest, beneficial for clinical decision-making processes.

**Weaknesses:**

1. While the paper uses 3 datasets, they are relatively simple. In this paper, authors use examples of medical images (fig 1) but did not actually experiment with any medical dataset. Using more diverse and complex real-life dataset (e.g., CIFAR-100 or MIMIC-CXR) would strengthen the paper and demonstrate broader applicability.
2. The paper lacks visualizations of counterfactuals. For example, when discussing improvements in counterfactual quality, it would be helpful to show visual comparisons with baselines. Providing real counterfactual comparisons would better substantiate the claims and help readers understand the model's advantages.

**Questions:**

Could you provide additional visual examples of counterfactuals generated by your model and baseline methods for each dataset? Visualizing these comparisons would be highly intuitive and beneficial for readers, helping them clearly see the improvements.

---

> ### Author Response · Authors · 2024-11-22
> **Response to Reviewer nCvr**
>
> ### Question: “visual examples of counterfactuals generated by your model and baseline methods for each dataset”
> **We are not interested in visual explanations, but we provided concepts counterfactual examples.** As explained in the paper, we are not interested in generating counterfactuals at the input level (e.g., images) because they are more difficult to interpret. Generating counterfactuals at the concept level also offers other advantages. It simplifies the information that should be contained in the latent representation ($z$ and $z{\prime}$), allowing the model to focus on salient information rather than encoding all the details necessary to recreate the full image. This makes it easier for the model to learn the latent distribution. Therefore, plugging a decoder into $z$ and $z{\prime}$ (which are related to concepts) to reconstruct the image with changed concepts would require a more complex latent space and significantly complicates the optimization process. However, we provided a few examples of counterfactuals at the concept level per model and dataset in the shared response.

---

> > ### Comment · Reviewer_nCvr · 2024-11-27
> >
> > Thank you for your explanation. It answers my question. And I would like to thank the authors to include the new working example on x-ray, it makes this paper much clearer.

---

### Official Review · Reviewer_hqCZ · 2024-11-03

**Soundness:** 2
**Presentation:** 3
**Contribution:** 2
**Rating:** 5
**Confidence:** 3

**Summary:**

The authors extend the concept bottleneck model framework to allow simulation of counterfactual concepts and labels. The method is based on a variational approach that models concepts as latent confounders, and approximates this posterior using concept labels at training time. This extension opens the door to explaining prediction at a finer level of detail, focusing on intermediate concepts rather than pixel values.

**Strengths:**

The authors tackle an extremely challenging problem (counterfactual reasoning using deep learning); making meaningful progress along this direction would have a big impact.

The authors show how training using a structured objective allows for test-time manipulations that shed light on how predictions are made

They extend CBM to include counterfactual sampling, which in principle allows a single model to estimate quantities at all three levels of Pearl’s ladder of causation.

The proposed method relies on concept labels, which may be difficult to collect in realistic settings.

**Weaknesses:**

The training objective is fairly complex and introduces many hyperparameters. The authors do not discuss how to tune these.

There is a disconnect between the working example of x-ray imaging and the types of datasets considered in the experiments.

I felt that details of the SCM design and subsequent training (e.g. ignoring p(x|z)) were not justified

**Questions:**

Additional comments/questions:
* [L042-044] This is a bold claim....what about the field of XAI? Pearl's ladder of causality has been quite influential in deep learning (including XAI), so I would assume that any methods aimed at counterfactual explanations would address these questions to some extent.
* I find the abstract a bit strange. Causal inference is a broad field attempting to answer (from a statistical perspective) the how and why questions, but it is not explicitly mentioned. Also, "existing CBMs" are mentioned in passing but the authors don’t say what they are or how they work. Maybe it would be better to more precisely situate the paper in the context of recent attempts to integrate causal inference with deep learning.
* [Sec 2] I don't insist, but it would be nice to see a citation acknowledging early efforts of using VAEs for causal inference [https://proceedings.neurips.cc/paper/2017/hash/94b5bde6de888ddf9cde6748ad2523d1-Abstract.html]
* [Eqn 1, L123] I don't understand why sampling a counterfactual concept c' does not also require sampling a fresh set of covariates x'. It seems to me that for concepts that matter in the real world, you would need to update the observed features as the concept is manipulated. To ground this in the x-ray example, if the concept is changed from "narrow joint space" to "bone spurs", would the distribution over pixels not also need to be updated? Clarifying this point seems important to justify why the $p(x|z)$ term is ignored in the optimization [L152].
* [L169] how are specific values for $\lambda_i$ [Table 6] chosen? Was a validation set used for tuning? Is the method sensitive to these values? I'm not satisfied with "these details are in the released code" [L850]
* [L260-265] The experiments inherit some datasets from prior papers, such as Koh et al 2020. Is there a reason why the OAI x-ray dataset from that paper was not used? X-rays are used as a working example throughout the paper [Fig 1, Fig 2, Sec 3.2] so this omission surprised me.
* [Sec 4, Sec 5] How does the method do when asked to simulate concept pairings that are OOD w.r.t. its training data? I would imagine for the CUB dataset there will be combinations of bird characteristics that are not represented by real-life birds; is it possible to examine the predictions for these concepts?. The strength of human counterfactual reasoning comes in part from OOD generalization, e.g. imagining a pink elephant. Replicating this using ML seems quite difficult. In the end I still have questions around whether what is being done here is meaningfully different from structured generative/discriminative modeling [L503-509] with interpretable features that can be adjusted on the fly. I think this is certainly related to counterfactual reasoning, but may not encompass it completely.
* [L516] If I express the counterfactual as E[Y|do(X)=X', Y=Y'] and the interventional as E[Y|do(X)=X'], then it would seem that counterfactual explainability methods can answer the "how" question by just marginalizing out the specific observation, i.e. E[Y|do(X)=X'] = E_{Y'}[ E[Y|do(X)=X', Y=Y']. You could replace the observation being a label (Y=Y') with the observation being a concept or other covariate as needed. Is there something wrong with this line of reasoning? Modeling counterfactuals seems more challenging than modeling interventions, so I am wondering if the authors have thought of how to go from one to the other in a post-hoc manner without designing a method that explicitly models all three levels of Pearls' ladder.

---

> ### Author Response · Authors · 2024-11-22
> **Response to Reviewer hqCZ (1/2)**
>
> ### The SCM is not well-detailed
> **1. Counterfactual generation in XAI has a different definition and assumptions w.r.t. Pearl’s causal framework.** In this respect, the counterfactual generation we refer to fits in the definition of Wachter’s framework [1] which does not require the explicit definition of an SCM (see L103-107).
>
> **2. In CBMs the “ground truth causal graph” of the SCM is the bipartite graph connecting concepts to tasks, as shown by the PGM in L118.** However, in the XAI framework for counterfactual generation it is not required to get an explicit set of structural equations in order to generate counterfactuals.
>
> ---
>
> ### “Any methods aimed at counterfactual explanations would address these questions to some extent.”
> **We emphasize that our method addresses these questions by design**, unlike most counterfactual methods, which are post-hoc. As Rudin [2] notes, the explanatory robustness of surrogate models might be questionable as they may not necessarily align with the original model’s decision-making process. Interpretable by-design architectures, such as CBMs and CF-CBMs, do not suffer from this issue. Our approach allows answering both “how” and “why not” questions directly, even on unstructured data—a significant challenge, particularly for “how” questions. For example, measuring the impact of modifying a single pixel often lacks meaningful interpretation, even with a counterfactual generator.
> While counterfactual explanations generally enable answering “why not” questions and provide some insights into “how,” they often fail to properly address the latter.
>
> ---
>
> ### Better contextualize the work in the abstract
>  **We modified it** in the reviewed version, thanks for the suggestion.
> We added the following sentence at L13: “While current approaches in
> causal representation learning and concept interpretability are designed to address
> some of these questions individually (such as Concept Bottleneck Models, which
> address both “what” and “how” questions), no current deep learning model is
> specifically built to answer all of them at the same time.”
>
> ---
>
> ### Citations on VAE for causal inference
> **Our model does not focus on causal inference**. Its goal is to provide insights into the model’s reasoning process rather than discovering or encoding causal relationships. However, we acknowledge the relevance of the cited works and similar research in adjacent areas and have included the following clarification at L48: “Causal inference aims to answer this question from a statistical perspective [3,4], seeking to discover and/or exploit the true causal relationships in the data. In contrast, we focus on addressing these questions from the model’s perspective, shedding light on its decision-making process. From this viewpoint, state-of-the-art counterfactual generators …”
>
> ---
>
> ### “Why sampling a counterfactual concept c' does not also require sampling a fresh set of covariates x' (why the  p(x|z)  term is ignored in the optimization [L152]?)”
> **This is an advantage rather than a limitation.** Since $z$ and $z{\prime}$ represent the latent distributions of the concepts $c$ and $c{\prime}$, there is no need to resample a fresh set of covariates $x{\prime}$. We are not focused on reconstructing or generating $x$ and $x{\prime}$, so the $p(x|z)$ term is ignored. This simplification allows the model to encode only the most salient information about the concepts, avoiding the need to capture all the details required to reconstruct the image. Ignoring $p(x|z)$ significantly reduces the complexity of the optimization process, making counterfactual generation on complex data, such as images, easier.  However, we thank the reviewer to have pointed this out and we will rephrase L152 as follows:
> “In practice, we assume that the input $x$ is always observed at test time, making the $p(x|z)$ term irrelevant. This allows $z$ and $z{\prime}$ to encode only the most salient information about the concepts $c$ and $c{\prime}$, rather than all the details needed to reconstruct $x$. As a result, the optimization process is significantly simplified, enabling efficient counterfactual generation for complex data like images.”
>
> ---
>
> [1] Wachter, Sandra, Brent Mittelstadt, and Chris Russell. "Counterfactual explanations without opening the black box: Automated decisions and the GDPR." Harv. JL & Tech. 31 (2017): 841.
>
> [2] Rudin, Cynthia. “Please Stop Explaining Black Box Models for High Stakes Decisions.” ArXiv abs/1811.10154 (2018): n. pag.
>
> [3] Louizos, Christos et al. “Causal Effect Inference with Deep Latent-Variable Models.” Neural Information Processing Systems (2017).
>
> [4] Xia, Kevin Muyuan et al. “Neural Causal Models for Counterfactual Identification and Estimation.” ArXiv abs/2210.00035 (2022): n. pag.

---

> > ### Comment · Reviewer_hqCZ · 2024-11-26
> > **thanks**
> >
> > I read the rebuttal and paper revisions. Clearly the authors have considered my feedback and put a lot of effort into the rebuttal, which I appreciate.
> >
> > I am still pondering the idea that while structure about the data (SCM) is not required for XAI counterfactuals, the authors do make structural assumptions about what is happening with the model predictions (e.g. PGM in Eqns 1 and 2). Could you say more about why you chose that factorization, and whether others were considered.
> >
> > I remain concerned about the hyperparameter tuning. I appreciate that the authors discuss the possibility of tuning various tradeoffs, but to be frank, the idea of tuning seven hyperpaameters over some mixture of three desirable metrics sounds like a practical headache.
> >
> > The new datasets are nice to see.

---

> > > ### Author Response · Authors · 2024-12-03
> > >
> > > Thank you for your comments.
> > >
> > > However, we would like to further clarify these last points.
> > >
> > > Regarding the need for a SCM:
> > >
> > > - Wachter et al. define counterfactual explanations as an optimization process aimed at altering the label with minimal modifications to the input. This definition does not inherently require or rely on the use of SCMs.
> > >
> > > - That said, if we were to frame the discussion in terms of SCMs, a CBM can be viewed as a “flat” SCM, where all concepts are causative of the label and are assumed to be independent of one another. This assumption forms the foundation of CBMs, and our work adopts the same perspective for consistency.
> > >
> > > Regarding our PGM:
> > >
> > > - The lower part of the PGM is based on the definition of CBMs, as outlined in Section 2. Concepts represent the generative factors of $x$ and $y$.
> > >
> > > - The upper part of the PGM extends this framework by considering concepts not only as the generative factors of $x$ and $y$ but also as the generative factors of counterfactuals ($c{\prime}$). This extension reflects the idea that concepts serve as the starting point and encapsulate the critical information leading to specific counterfactuals, still viewed through the lens of an optimization process.

---

> ### Author Response · Authors · 2024-11-22
> **Response to Reviewer hqCZ (2/2)**
>
> ### Model’s behavior on OOD counterfactuals
> **This question is more closely related to the “how” questions in XAI.** In the context of XAI, counterfactual generation is conditioned on a different desired label, requiring the model to produce a new input (concepts in this case) that leads to the target label. This differs from counterfactuals in causal inference, where a feature is manually modified (e.g., making an elephant pink) to observe its effects on other features. While these two paradigms are related, they serve different purposes. What you describe aligns more with manually modifying specific concepts (e.g., attributes of a bird) to create an OOD sample and observing how the model predicts it. This approach is indeed related to “how” questions and allows users to inspect the model’s behaviour under different scenarios. In contrast, counterfactuals in XAI are generally expected to remain in-distribution, as they represent feasible states leading to the desired label [1]. Notably, from our experiments, we observed that our counterfactuals, leveraging VAE-like methods, tend to remain in-distribution even when the input is OOD.
>
> ----
>
> ### Answer how questions exploiting counterfactual information
> We agree that it is possible to answer “how” questions in the way you described, but this approach is inefficient. Generating multiple counterfactuals by iterating over all possible $y{\prime}$ values provides only general information about the impact of specific features or pixels on a label. It does not enable a fine-grained analysis of how modifying a specific feature to a particular value affects the prediction. Moreover, these measured impacts depend on what the counterfactual generator has encoded, which may not fully align with what the model has learned, leading to potential discrepancies. Developing a method to address all the three questions without additional components is quite challenging. One possibility to generate counterfactuals exploiting the how answers is to iteratively modify concepts based on their measured impact (e.g., using CaCE) until the label reaches the desired value. However, this approach often results in OOD counterfactuals, which are undesirable [1].
> Answering “how” questions starting from the generated counterfactuals remains a difficult task. We find this to be an interesting direction for future work, potentially exploring ideas along the lines of what the reviewer suggested. We appreciate the valuable feedback and will consider it in our future research.
>
> ----
>
> [1] Wachter, Sandra, Brent Mittelstadt, and Chris Russell. "Counterfactual explanations without opening the black box: Automated decisions and the GDPR." Harv. JL & Tech. 31 (2017): 841.

---

### Author Response · Authors · 2024-11-22
**Answer to all reviewers and ACs (1/2)**

We thank the reviewers for their insightful feedback. We are pleased to see that reviewers recognized the significance of our work in addressing an important problem with potential value for real-life applications (hqCZ, nCvr), appreciated the straightforward and intuitive nature of our model (nCvr), and acknowledged our contribution as both novel and important (nCvr, RbDp). Additionally, we are glad that reviewers found our paper to be well-written and easy to read (nCvr, 9mnf, RbDp) and noted the solid evidence supporting the effectiveness of our approach (nCvr, 9mnf). Furthermore, we are thankful for all the suggestions the reviewers made. These have certainly improved the quality of our manuscript, and we hope we can address your concerns here.

Below, we reply to questions shared by two or more reviewers.  We reply to specific questions reviewers had in comments under their respective reviews.

## Summary of Changes
In our revised manuscript, we have included the following changes (in purple in the revised paper) to take into account the reviewers’ feedback:
- Replaced all x-ray knee examples with x-ray lung examples to align them with the experimental settings (updated in the Figure 1, Figure 2, Sections 1, 2, 3.1, and 3.3).
- Added two datasets (CIFAR10 and SIIM Pneumothorax) to expand the experimental settings with real-world datasets lacking concept annotations, further demonstrating the applicability of our method in such cases (Section 5), without altering the overall conclusions.
- Moved the discussion on Time Efficiency from Section 5 to Appendix D to make room for the newly added results.
- Included a discussion on the meaning of the introduced hyperparameters in Appendix C.3, along with an additional ablation study examining their impact.
- Added examples of generated counterfactuals in Appendix E.
- Enhanced the contextualization of our method in the Abstract.
- Rephrased and clarified several points to address specific concerns raised by the reviewers.
- Included a few missing references.

## Shared Answers

### Working example on x-ray
We thank the reviewers for the suggestion. The initial X-ray example was based on the Osteoarthritis Initiative (OAI) dataset [1], which is not publicly available. Obtaining the required permissions to use it would have delayed the experiments significantly.
In the meantime, **we conducted additional experiments on the SIIM Pneumothorax dataset** [2], which also contains x-ray scans, focusing on detecting Pneumothorax. We updated all examples and abstract figures in the paper to reflect this new dataset instead of the OAI dataset.
We appreciate the reviewer’s feedback, which has helped make the paper clearer and more aligned. (Please see the next response for details on the experiment conducted with the SIIM Pneumothorax dataset.)

----

### “(..) the proposed method relies on concept labels (..)”
**Concept-supervision is a key feature of CBMs** As CBMs are designed within the explainable AI domain, concept supervision should not be viewed as a limitation but rather as a key feature enabling alignment with the intended target of the explanations (e.g. human). In scenarios where concept supervision is unavailable, proxy models can be employed, often leveraging pretraining strategies such as CLIP. To demonstrate this, we have included additional results in Section 5, showing that our method achieves similar conclusions to the one presented in the main paper on the real-world CIFAR10 and SIIM Pneumothorax dataset, where concepts were extracted following [3].

----


[1] OsteoArthritis Initiative, http://doi.org/10.17616/R3RP9X

[2] Anna Zawacki, Carol Wu, George Shih, Julia Elliott, Mikhail Fomitchev, Mohannad Hussain, ParasLakhani, Phil Culliton, and Shunxing Bao. SIIM-ACR Pneumothorax Segmentation. https://kaggle.com/competitions/siim-acr-pneumothorax-segmentation, 2019. Kaggle.

[3] Oikarinen, Tuomas P. et al. “Label-Free Concept Bottleneck Models.” ICLR (2023).

---

> ### Author Response · Authors · 2024-11-22
> **Answer to all reviewers and ACs (2/2)**
>
> ### Discussion on hyperparameters in the training objective
> The hyperparameters in the loss ($\lambda$) were chosen through validation on a subset of the training to achieve the best possible trade-off across all metrics important for counterfactual evaluation, such as validity, proximity, and sparsity, for both our method and the baselines. While our method introduces more hyperparameters, it provides greater flexibility to specify the desired objective. As counterfactual metrics often conflict, creating inherent trade-offs (as shown in our experiments), there is no one-size-fits-all solution. Thus, optimizing the counterfactual generator based on the metric of interest is crucial, and our method facilitates this.
> Each hyperparameter in the loss has a specific role in steering the model toward a particular behavior:
> - $\lambda_1$ (task-related) and $\lambda_2$ (concept-related) are important in all configurations as it remains the main goal of the model.
> - $\lambda_3$ prioritizes validity, encouraging counterfactuals with predictions matching $y{\prime}$, though potentially compromising proximity or sparsity.
> - $\lambda_4$ and $\lambda_5$ emphasize generating more realistic counterfactuals, potentially reducing proximity, which may trade off against validity and sparsity.
> - $\lambda_6$ and $\lambda_7$ promote sparsity by reducing the number of changes from the input, which may trade off against validity. However, we observed that $\lambda_6$ has minimal influence on the optimization process, with $\lambda_7$ being the key hyperparameter driving this behavior.
>
> This flexibility makes our model more versatile and practical compared to baselines, which have limited or partial support for these trade-offs (e.g., VAECF has the possibility to be flexible on validity).
> We appreciate the reviewers’ feedback and we clarified how to select these hyperparameters in the revised manuscript in Appendix C.3. For additional evidence, we conducted an ablation study on the MNIST-Add dataset, modifying $\lambda_3$, $\lambda_4$, $\lambda_5$, and $\lambda_7$ to evaluate the model’s behavior across different objectives:
>
>
> | Objective Focus           | λ₃   | λ₄   & λ₅    | λ₇    | Validity (%)     | Sparsity (%)      | Proximity (%)    |
> |---------------------------|------|------------|-------|------------------|-----------------|-------------------|
> | Higher focus on validity  | 0.4  | 2    | 0.55 | 99.1 ± 0.2       | 19.0 ± 0.4    | 4.3 ± 0.7     |
> | Higher focus on proximity | 0.2  | 4    | 0.55  | 93.6 ± 1.6       | 14.2 ± 0.4    | 3.6 ± 0.0     |
> | Higher focus on sparsity  | 0.2  | 2    | 0.70   | 74.4 ± 0.01      | 6.1 ± 0.4    | 2.6 ± 0.0     |
>
> ----
>
> ### Show examples of cf in the input space
>
> **Our model produces counterfactual explanations only within the concept space**. Like other deep learning classifiers, it does not model the input space distribution on which it operates conditionally (i.e., discriminatively). This approach is intentional, as we argue that concept-based counterfactuals align more closely with human understanding and are more actionable. For instance, while minor pixel changes in an X-ray offer limited value, flipping a concept such as "Visible Pleura Line" provides meaningful and interpretable insights to the physician. However, as you pointed out, it would be helpful to see a comparison of the counterfactuals generated by the different methods across the datasets, so we have included these examples in the revised version of the manuscript in Appendix E, where it is visible that our counterfactuals are also qualitatively better than the other baselines.
>
> ----

---

### Author Response · Authors · 2024-12-03
**Thanks**

We would like to thank all the reviewers for the comments and for helping us improve the paper.

---

### Meta-Review · Area_Chair_PyAY · 2024-12-17

**Metareview:**

The paper extends the concept bottleneck model (CBM) framework to enable counterfactual reasoning, allowing for finer-grained explanations of model predictions using intermediate concepts instead of raw pixel values. The proposed method relies on a variational approach that treats concepts as latent confounders, with posterior distributions approximated using concept labels during training. This approach addresses the "what," "how," and "why not" questions about predictions, offering a unified method for interpretability and intervention in predictions.

Strengths

+ The paper addresses a challenging and impactful problem: integrating counterfactual reasoning with deep learning models for enhanced interpretability.
+ The method enhances traditional CBMs to handle counterfactual sampling, potentially addressing all three levels of Pearl's ladder of causation.
+ Solid experimental results demonstrate that the proposed method maintains classification performance comparable to standard CBMs and black-box models.
+ The paper is well-organized, clearly written, and effectively introduces the motivation, contributions, and experimental findings.
+ The approach improves the interpretability of counterfactuals, which is valuable for real-world applications, particularly in domains like medical decision-making.
+ The method is tested against multiple baselines and datasets, showing robustness in various settings.

Weaknesses

+ The training process introduces multiple hyperparameters, but the authors do not provide sufficient details on tuning or sensitivity analysis, raising concerns about practical usability.
+ Despite using medical imagery as a motivating example, no experiments were conducted on medical datasets, limiting the method's demonstrated applicability to real-world scenarios.
+ The datasets used (e.g., dSprites, MNIST, and CUB) are relatively simple, and testing on more complex or diverse datasets (e.g., MIMIC-CXR or CIFAR-100) would strengthen the claims.
+ The method relies on handcrafted concepts, which may be difficult and labor-intensive to define in practice, and predictive performance depends on the quality of these concepts.
+ The lack of visualizations comparing counterfactuals generated by the proposed method and baselines weakens the qualitative evaluation of the approach.
+ Concerns remain regarding the interpretability trade-offs introduced by additional black-box components and the absence of justification for certain design choices (e.g., ignoring p(x∣z)p(x|z)p(x∣z)).


Most concerns have been addressed by the authors during the rebuttal period.

**Additional Comments On Reviewer Discussion:**

This paper started with slightly positive ratings. After rebuttal, most reviewers are satisfied with the response, some of them raising the scores. The final ratings are 8, 8, 6, 5. The only negative reviewer’s major concern is the involvement of 7 hyperparameters during training. I tend to agree. Given the overall contribution of the paper, I would still recommend acceptance. However, I would strongly encourage the authors to release the code to improve reproducibility given the hyperparameter issue.

---

### Decision · Program_Chairs · 2025-01-22

Accept (Poster)